Resource

# DynamicAtlas: a morphodynamic atlas for *Drosophila* development

Matthew F. Lefebvre [1,10], Vishank Jain-Sharma [1,10], Nikolas Claussen[1,10], Noah P. Mitchell [1,2,9,10], Marion K. Raich [3], Hannah J. Gustafson[1,4], Friederike E. Streichan[5], Andreas R. Bausch [3,6,7,8] & Sebastian J. Streichan [1,4] ✉

Living organisms develop their shape through the interplay of gene expression and mechanics. While atlases of static samples characterize cell fates and gene regulation, understanding dynamic shape changes requires live imaging. Here we present DynamicAtlas: a 'morphodynamic atlas' of live and static datasets from 500 *Drosophila melanogaster* embryos (wild type and 18 mutants), aligned to a common morphological timeline. Surprisingly, characterizing wild-type surface tissue flows reveals distinct 'morphodynamic modules'—time periods in which the global pattern of motion is stationary—corresponding to key developmental stages. Mutant analysis shows stationary flow patterns depend on genes that break spatial symmetry along the dorsal–ventral axis. Temperature perturbations indicate that morphodynamic modules change in response to accumulated tissue deformation, rather than elapsed time. Extending our approach to the embryonic *Drosophila* midgut, we find modules in covariant measures of the dynamic three-dimensional surface. DynamicAtlas provides a high-resolution framework for studying shape formation across living systems.

There is a strong interest in constructing transcriptomic and protein expression atlases to tackle outstanding questions in morphogenesis[1–7]. By necessity, existing atlases have been static representations of embryo components at a collection of specific points in time[4,8,9]. Many atlases map single-cell RNA sequencing (scRNA-seq) data, by classifying cell fates from clusters in transcriptomic space[2,4,7]. These bottom-up approaches have illuminated new connections between disparate components of the morphogenetic program[2,4,5,7,10]. However, despite reconstructive techniques, scRNA-seq experiments do not fully preserve spatiotemporal information[11]. This demands new methods for registering independent datasets in both space and time[12].

A notable spatial registration technique was developed by the Berkeley *Drosophila* Transcription Network Project atlas (BDTNP atlas), which measured three-dimensional (3D) gene expression patterns in the *Drosophila melanogaster* blastoderm from fixed (static) samples[1]. Embryos were coarsely binned in time, and spatially registered using costained pair-rule gene (PRG) expression patterns. The BDTNP atlas has facilitated broad discovery of the gene regulatory networks underpinning morphogenesis[13,14].

[1]Department of Physics, University of California Santa Barbara, Santa Barbara, CA, USA. [2]Kavli Institute for Theoretical Physics, University of California Santa Barbara, Santa Barbara, CA, USA. [3]Center for Protein Assemblies (CPA) and Lehrstuhl für Biophysik (E27), Physics Department, Technical University of Munich, Garching, Germany. [4]Interdisciplinary Program in Quantitative Biosciences, University of California Santa Barbara, Santa Barbara, CA, USA. [5]Independent Researcher, Santa Barbara, CA, USA. [6]Technical University of Munich, TUM School of Natural Sciences, Department of Bioscience, Heinz Nixdorf Chair in Biophysical Engineering of Living Matter, Garching, Germany. [7]Matter to Life Program, Max Planck School, Munich, Germany. [8]Center for Organoid Systems and Tissue Engineering (COS), Technical University of Munich, Garching, Germany. [9]Present address: Department of Molecular Genetics and Cell Biology, The University of Chicago, Chicago, IL, USA. [10]These authors contributed equally: Matthew F. Lefebvre, Vishank Jain-Sharma, Nikolas Claussen, Noah P. Mitchell. ✉e-mail: streicha@ucsb.edu

Characterizing morphogenetic dynamics from atlases remains challenging. For example, genes are thought to be important drivers of morphogenesis[15]. In the BDTNP atlas, gene expression patterns are only sparsely mapped in time[1], and the relationship between genetic patterns and tissue deformation patterns remains elusive. To understand how gene expression patterns in an animal regulate its shape changes, global tissue motions must be observed directly. This requires a 'morphodynamic' atlas that incorporates live, in toto videos.

A major hurdle in constructing a morphodynamic atlas is to integrate spatiotemporal information from different classes of data into a common framework. This requires three computational steps: (1) spatial alignment between experiments, (2) temporal alignment between experiments and (3) construction of a single morphogenetic timeline across all experiments.

In this resource, DynamicAtlas, we address this challenge by generating a dynamic protein expression atlas that integrates videos spanning much of *Drosophila* embryogenesis. Our atlas comprises live and fixed datasets from 500 unique embryos, with wild type (WT) and 18 mutant genotypes (detailed in Supplementary Table 1). All data were captured using in toto multiview light-sheet microscopy, enabling global analysis of dynamics[16–18]. The atlas defines spatial coordinates using tissue cartography, and temporal coordinates using tissue morphology. Datasets are publicly accessible on a Dryad repository[19].

We have developed two open-source software interfaces to interact with DynamicAtlas ('Code availability'). Users can explore existing resources with a Python-based Jupyter notebook interface[20], and can computationally integrate new data into DynamicAtlas with our MATLAB-based interface, which contains methods for de novo timeline creation and fixed data timestamping[21]. Our resource is specifically designed to incorporate future contributions.

We demonstrate how to use the DynamicAtlas by characterizing kinematics of morphogenesis in *Drosophila*. First, we use an ensemble of WT videos to characterize global surface tissue flows during early embryo development. We then construct additional ensembles to study salient features of these flows under genetic and temperature perturbations. Finally, we construct a morphological timeline to study deformations of the *Drosophila* midgut, illustrating the applicability of our approach to complex, dynamic 3D shapes.

## Results

### Construction of a morphodynamic atlas of *Drosophila* development

In DynamicAtlas, we integrate independent classes of experimental data onto a fixed spatial reference frame, indexed along a single morphological timeline (Fig. 1). Both live and fixed expression data are visualized using a cartographic approach that makes two-dimensional (2D) projections of the embryo's 3D surface (Fig. 1a,b). By imaging samples in toto, we can perform spatial alignment using the full embryo geometry (Fig. 1c). Our software automatically aligns a cartographic projection of the 3D embryo surface with the embryo body axes, using techniques previously developed in refs. 22 and 23 (Methods section 'Tissue cartography'). This method enables rapid analysis of tissue dynamics and expression patterns across embryos, at subcellular resolution.

Our alignment-and-comparison technique enables description of the dynamic morphogenetic program in terms of underlying physical fields in space. Single-cell comparison across embryos is complicated by positional variations across samples. Instead, we extract meso-scale information by smoothing gene expression patterns, velocity fields and measures of anisotropy. These spatially aligned fields are ripe for quantitative hypothesis testing, as they can be compared across the embryo's body plan. Recently, this biophysical field-theoretic approach has led to quantitative models relating mechanical force, gene expression and tissue geometry[22,24–28].

After spatial alignment is performed, recordings must also be temporally aligned (Fig. 1d) while accounting for natural variations in morphogenetic rates. This raises two central questions: what is the relevant timeline for analysis of morphogenetic processes, and how can disparate timelines be consolidated? Canonical morphological milestones have been used to define discrete stages to track developmental progress. Here we extend this technique to high temporal resolution by using continuously deforming geometrical features of the tissue to create a 'morphological time' variable that is suitable for quantitative analysis. Figure 1e depicts a unified morphological timeline, with distinct embryos labeled for the PRGs in Fig. 1b overlaid at three sample timepoints along the timeline (generated using tools in Fig. 1f).

We proceed in two steps: (1) timelines of individual live datasets are dilated to align along a consensus timeline (section 'Aligning live datasets based on PRG expression') and (2) fixed samples are placed appropriately in the consensus timeline (section 'Aligning fixed datasets').

In general, for alignment of live videos, we first select a suitable morphological feature that allows us to score the similarity of two images (Fig. 2a(i)–(iii)). Rather than performing a global 'rigid alignment' by a fixed temporal offset, we instead compare all pairs of frames between videos, creating a rectangular matrix of correlations (Fig. 2b(i),(ii)). A fast-marching algorithm then finds the optimal path through this matrix, matching every timepoint of one video to its best match in the other video while respecting temporal order, thereby naturally accounting for their variable morphogenetic rates (Fig. 2b(i)–(iii)).

**Aligning live datasets based on PRG expression.** As shown in Fig. 2a(i),(ii), the PRG Runt is expressed in stripes along the embryonic dorsal–ventral (DV) axis that deform as the epithelium flows. By imaging a collection of embryos expressing a llama tag for Runt[29], we extract the continuously deforming morphology of Runt stripes as a landmark for morphological timing to carry out steps (1) and (2) above (Fig. 2a(iii)). We thereby leverage live datasets of Runt expression to build a consensus morphology of Runt stripe geometry. By linking the timelines for each pair of Runt embryos using our fast-marching algorithm (Methods, section 'Fast-marching algorithm'), we generate a single, consensus timeline.

**Aligning fixed datasets.** As shown in Fig. 2c(i), we designed our atlas such that all fixed embryos are multiply immunostained against both Runt and a second gene product target of interest. For each embryo, we extract the static geometry of Runt stripes to correlate its position against the consensus Runt stripe shapes from live datasets (Fig. 2c(ii)), thereby determining the timestamp at which the fit residual is minimal (Fig. 2c(iii)). This process uses stripe geometry as a stopwatch against which we timestamp all fixed samples in the atlas. Any costained targets of interest are therefore aligned in time based on the position of the Runt stripe. For our consensus timeline based on live Runt nanobody data ($n = 5$), we find that this method timestamps the average fixed sample to the timeline with a 1-$\sigma$ uncertainty of ±2 minutes (Methods section 'Uncertainty of morphological timestamping'). For other ensembles, the timestamping uncertainty can be similarly determined empirically. In general, an ensemble requires a minimum of three live samples to perform the timestamping, and the precision of this process can be increased to a desired level by increasing the number of live samples recorded.

While the capacity to align live and fixed datasets based on a shared protein expression feature is valuable, Runt expression data are not present in all live imaging datasets. To integrate these other types of live data, we therefore need to extend our morphological approach.

**Aligning live datasets based on tissue deformation.** Embryonic tissue deforms during morphogenesis, and its instantaneous rate of deformation can be captured by flow fields[22] (Fig. 2d(i)). The degree of total deformation can serve as a benchmark for defining morphological time, and can be reconstructed from the instantaneous flow fields (Supplementary Note 5).

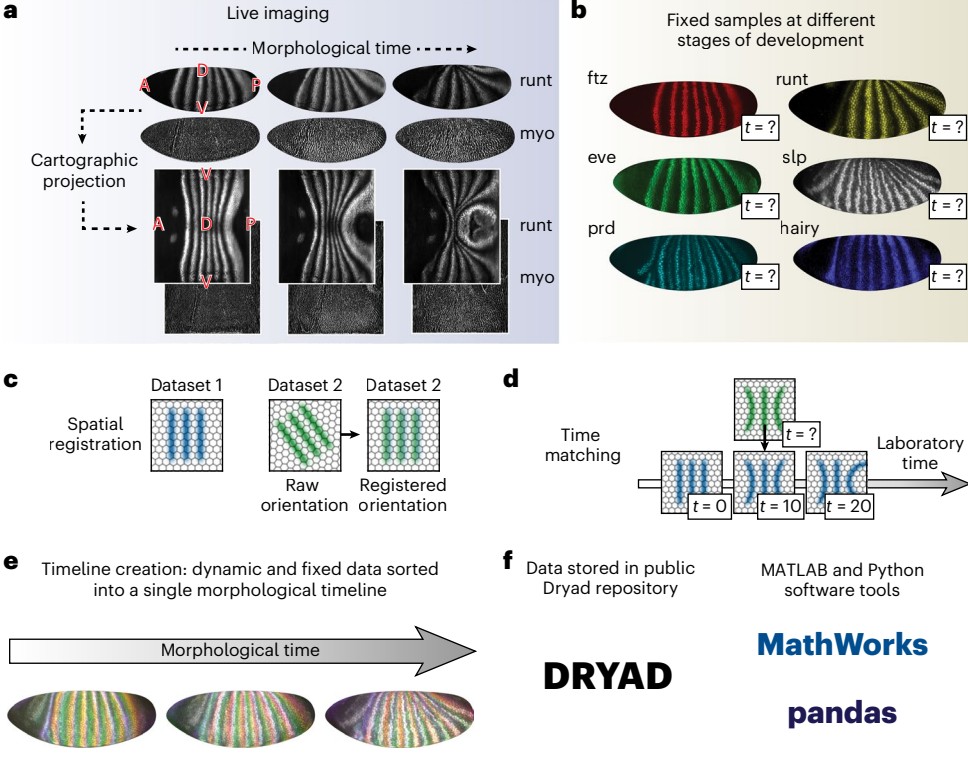

**Fig. 1 | DynamicAtlas enables spatiotemporal alignment of live and fixed datasets onto a common morphological timeline. a**, Live light-sheet imaging generates in toto volumetric datasets, which are then spatially registered into a fixed 2D cartographic parameterization. Representative snapshots from live datasets of a Runt nanobody reporter (runt) and a myosin marker (myo) are shown: 3D surface representations (above) and 2D cartographic projections (below). Anterior is to the left. 3D representations: dorsal is up. 2D projections: dorsal is in the center, ventral is on the top and bottom. **b**, Fixed embryos require a method for generating timestamps. Representative fixed samples stained for the PRGs: ftz, runt, eve, slp, prd, hairy. **c**, Samples are registered to each other in space. **d**, Samples are registered to each other in time, by maximizing the similarity of a deterministic morphological feature that reproducibly changes with time. **e**, Schematic of a single morphological timeline after spatial and temporal alignment has been performed. Sets of six fixed samples, each stained for a distinct PRG from **b**, are overlaid at three sample timepoints along a WT morphological timeline. **f**, User interaction with the atlas: data are stored on a public Dryad repository, and can be explored with tools developed in MATLAB or Python. Python workflow uses the pandas library[56,57] to read properties of atlas datasets from reference spreadsheets.

We developed a method to compare integrated flow patterns, cross-correlating displacements of the tissue to one another using the same fast-marching technique as in aligning live imaging of Runt stripes. In both cases, the morphology of the tissue marks its placement in the morphological timeline. We fix the reference timepoint of the integration as the time when germband extension (GBE) tissue flow starts to rise substantially (Supplementary Note 6). As shown in Fig. 2d(i)–(iii) and Extended Data Fig. 1, this approach leads to aligned tissue morphology.

Note that we use total tissue deformation, and not the instantaneous flow field, as a developmental landmark. In many contexts, instantaneous flow fields can be relatively constant in time (for example, cells migrating with fixed speed), making them unsuitable as timestamps.

This method time aligns distinct embryos from tissue deformation alone, and does not require analysis of gene expression patterns. It is possible that this method could be used in other living systems.

## Morphogenesis follows a sequence of stationary flow patterns

Expressing tissue motion as a time-dependent velocity field defined over the surface of the embryo allows us to compare tissue motion at different times or in different embryos. For example, we can directly compute the difference of two velocity fields.

We measured instantaneous surface tissue flow (Fig. 3a) in all live datasets of the atlas using particle image velocimetry (PIV)[30]. A key observable from the instantaneous flow is the flow pattern (that is, flow normalized by its magnitude), which describes the directions cells move in from one moment to the next (Supplementary Note 5). When we examined these flow patterns in WT embryos, we observed

periods of time in which the global pattern of tissue velocity remains stationary (Fig. 3b). We stress that the tissue itself is not stationary, but instead the pattern of motion is stationary: although the cells are moving across the embryo, the flow pattern is stationary during certain discrete stages of development. (We call these flows 'quasi-stationary'— as their magnitudes may vary—and call the flow patterns 'stationary'.)

We quantified this observation by comparing the velocity pattern at time $t$ to the velocity pattern at another time $t'$, which defines a pairwise correlation function of tissue flow to itself. Computing pairwise correlations between flow patterns in time yields an autocorrelation matrix. We report the autocorrelation on a scale of −1 (perfectly anti-correlated) to 1 (perfectly correlated), shown in Fig. 3b for a video of a WT embryo, recorded from the onset of gastrulation to dorsal closure. The magnitude of autocorrelation was high during discrete blocks of morphogenetic time, irrespective of the correlation measure used (Extended Data Fig. 2), indicating that a modular sequence of flow patterns governs embryogenesis. We refer to these blocks as modules of stationary flow patterns. These 'stationary modules' correspond to established developmental episodes: (1) dorsal contraction (DC), (2) ventral furrow formation (VF), (3) GBE and, after about 2 hours of low correlation, (4) germband retraction (GBR).

Figure 3c shows the rapid consecutive transitions between the DC, VF and GBE modules. Each flow pattern is associated with cellular-level drivers. For example, a DV asymmetry in the basal myosin pool causes a dorsalward cell flow during the DC module[22]. Likewise, apical constriction of ventral cells drives cell flow to the ventral side during the VF module[31]. T1 transitions drive a dorsalward flow during the GBE module[32].

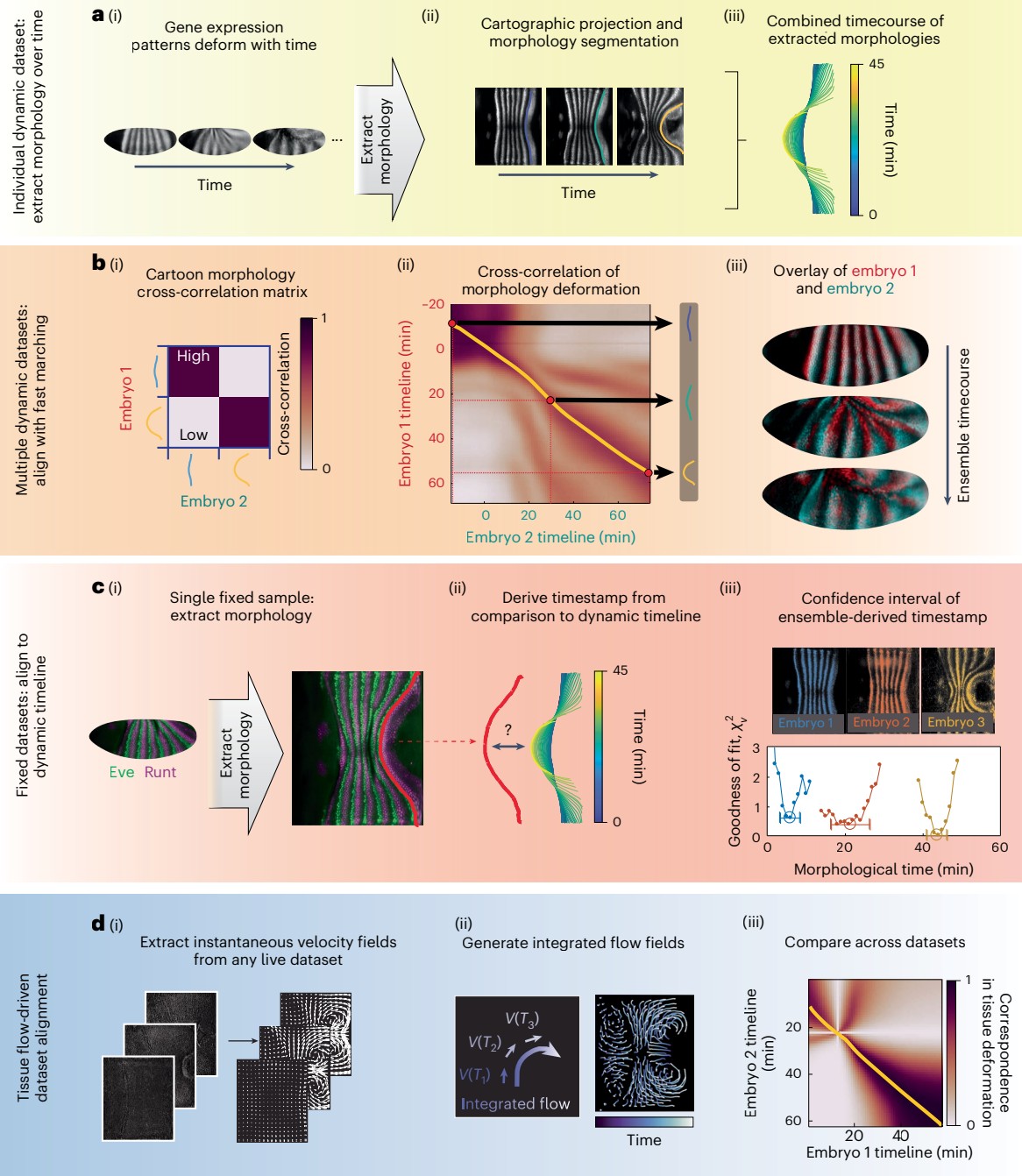

**Fig. 2 | Methods for aligning datasets in morphological time. a**, (i) Snapshots of a Runt nanobody live dataset. (ii) 2D projections of the snapshots. Runt stripe 7 shown in color. (iii) Overlaid stripe boundary morphologies for the full live dataset. Time 0 min: onset of GBE. **b**, (i) Cross-correlation matrix for two Runt stripe 7 morphologies. Correlation measure: Pearson correlation coefficient. (ii) Cross-correlation matrix between two live Runt nanobody datasets, using the measure in (i). Orange curve: optimal fast-marching correspondence curve. Three sample points in red. Red dotted lines: pairwise timeline correspondences from the points. Arrows indicate corresponding Runt stripe morphologies. Time 0 min: onset of GBE. (iii) Runt expression pattern overlays at the correspondence times in (ii). **c**, (i) Fixed sample costained for Eve (green) and Runt (purple). Runt stripe 7 shown in red. (ii) Comparing a single stripe morphology to the timecourse enables fixed dataset timestamping. (iii) Timestamping three fixed embryos costained for Runt onto the morphological timeline. Each colored curve shows the morphological comparison of one fixed sample to many timepoints along the consensus timeline. Circled points: timestamps. Error bars: 1-$\sigma$ uncertainty. Time 0 min: onset of GBE. **d**, (i) Snapshots of a live dataset labeled for a myosin marker (left), and corresponding instantaneous velocity fields (white arrows, right). (ii) Integrating instantaneous flow fields (left) to obtain cumulative flow trajectories (right). Colorbar: passage of time. (iii) Cross-correlation between two live datasets, based on flow-derived tissue deformation. Orange curve: correspondence curve between embryos, same method as **b**(ii). Time axes: video acquisition times.

While each individual embryo exhibited a nearly constant flow pattern during GBE, we wondered about sample-to-sample variability. Extending the correlation analysis to investigate cross-correlations between embryos, we observed remarkable similarity in the pattern of tissue flow between different samples, with the cross-correlation matrix of an example pair shown in Fig. 3d.

We used the atlas to further quantify how morphogenesis in individual embryos compares to an ensemble of embryos. To this

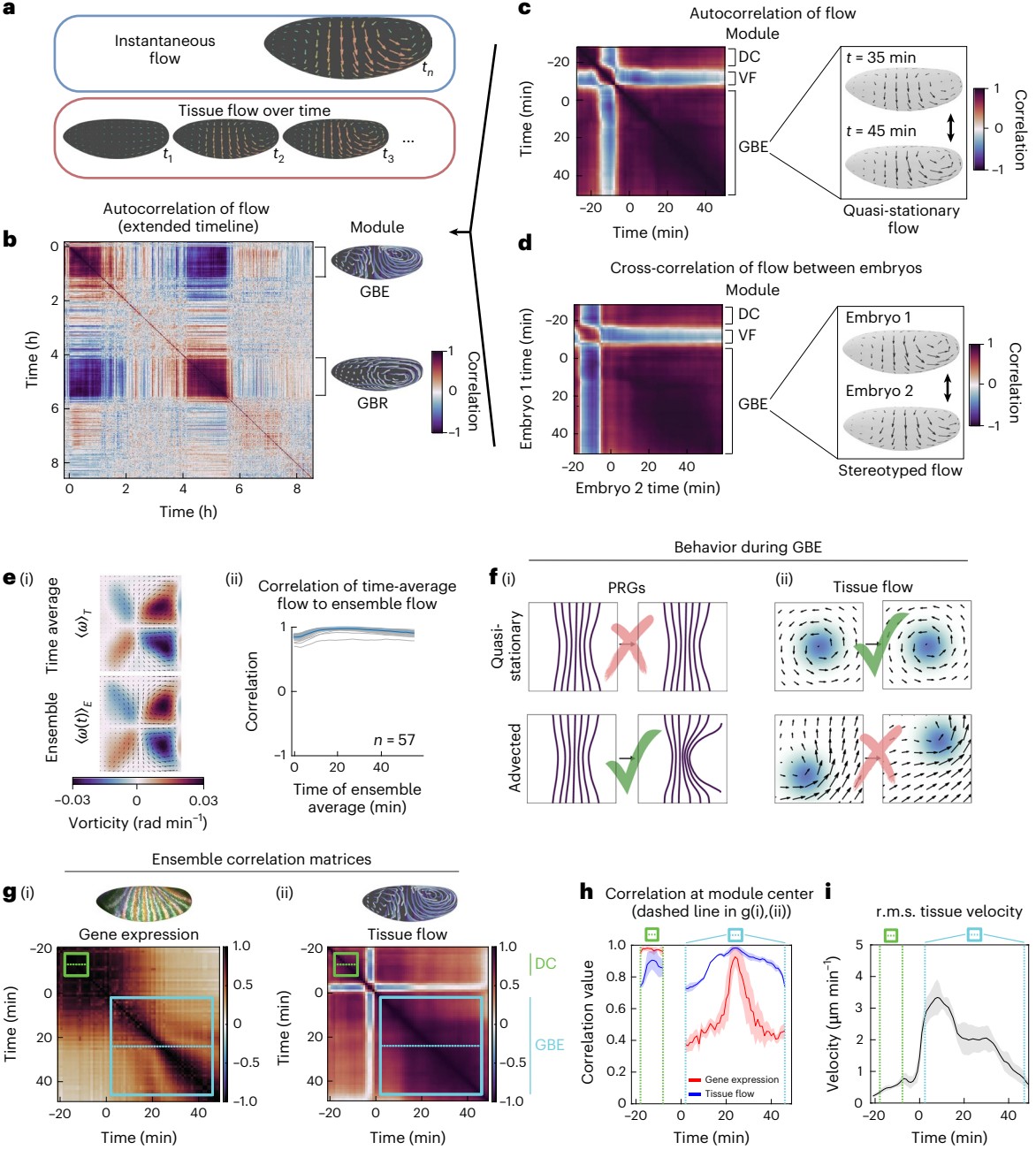

**Fig. 3 | Tissue dynamics exhibit reproducible modules of quasi-stationary flow.** Analysis of WT ensembles. Flow correlations: vorticity method. Time 0 min, hours: onset of GBE. **a**, Instantaneous flow: single timepoint (top), many timepoints (bottom). **b**, Autocorrelation matrix of a WT embryo, from onset of gastrulation to dorsal closure. Stationary modules for GBE and GBR labeled. Insets: integrated flow trajectories. **c**, Autocorrelation during gastrulation reveals three stationary flow modules: DC, VF and GBE. Inset: sample GBE flows. **d**, Cross-correlation during gastrulation. Inset: sample GBE flows. **e**, WT ensemble ($n = 57$). (i) Time-averaged flow of a sample embryo (top). Ensemble-averaged flow at a sample timepoint (bottom). Color overlay: vorticity. (ii) Correlations of individual time-averaged flows to the ensemble-averaged flow.

Blue curve: mean (curve time average: 0.94). Blue shading: s.d. **f**, (i) During GBE, PRGs are advected over time. (ii) By contrast, tissue flows are quasi-stationary (vorticity in blue). **g**, Auto-correlations of PRGs and corresponding flows, on a common ensemble timeline. Colored boxes: DC (green) and GBE (cyan) modules. Dashed lines: module centers. (i) Average autocorrelation matrix across six PRGs. Aligned ensemble: live samples ($n = 5$, Runt nanobody) and fixed samples ($n = 116$ total: 19 ftz, 42 runt, 13 eve, 8 hairy, 19 prd, 15 slp). Correlation measure: Pearson correlation coefficient. (ii) Average autocorrelation matrix of flow for a live ensemble ($n = 5$, labeled with Histone-RFP). **h**, Correlation values on the dashed lines in **g**(i),(ii). Shading: s.d. within ±2 min of the dashed line. **i**, r.m.s. surface velocity of the **g**(ii) ensemble. Shading: s.d.

end, we constructed two distinct averages within the GBE module (Fig. 3e(i)). First, we time-averaged flow. $\langle \omega \rangle_T$ represents the average flow pattern of GBE for an individual embryo (Fig. 3e(i), top). Second, we averaged many individual recordings into a single ensemble-average video. $\langle \omega(t) \rangle_E$ represents the ensemble-averaged GBE flow pattern of 57 spatiotemporally aligned WT embryos (Fig. 3e(i), bottom). The

correlation between time-averaged flow $\langle \omega \rangle_T$ and ensemble-averaged flow $\langle \omega(t) \rangle_E$ provides a correlation curve for each embryo during GBE. We found very high correlation for nearly all of GBE (Fig. 3e(ii)), with a time-averaged mean correlation value of 0.94 across the curve (vorticity correlation measure used, Methods section 'Correlations of velocity fields'). We conclude that not only is the GBE module stationary,

but also that its flow pattern is stereotyped across embryos. Embryos exhibit the same constant flow pattern during GBE.

To validate our PIV-based approach, we tracked 6,000 cells in an embryo with a cell membrane marker (CAAX-mCherry, Supplementary Video 3 and Supplementary Note 4) and derived the kinematics from single-cell trajectories. As shown in Extended Data Fig. 3, the kinematics based on cell trajectories differed little from the PIV measures.

PRGs are important contributors to GBE[15]. Yet, the in toto pattern of PRGs has only been explored in fixed samples. The atlas enables quantitative analysis of PRG pattern kinematics, shown for a set of six PRGs in Fig. 3f–h. Using the middle times for the DC and GBE modules, we quantified the autocorrelation patterns of both PRGs and flows (Fig. 3g(i),(ii),h). Correlations of PRGs and flows were both high during DC. In contrast, during GBE, PRG autocorrelation dropped sharply within 2 minutes. This fast PRG autocorrelation drop can be quantitatively explained from high vorticity (Fig. 3f(ii)) and speed increase (Fig. 3i) during GBE. PRG expression becomes rapidly reshaped as tissue vorticity increases, and within 2 minutes stripes no longer overlap faithfully (Methods section 'PRG decorrelation'). This fast decorrelation from flow suggests that PRGs no longer directly instruct GBE shortly after the stationary flow pattern has been established.

### DV patterning is a prerequisite for quasi-stationary flows

Mutant analysis often characterizes a phenotype in terms of the final tissue shape. The atlas introduces higher resolution quantitative methods to this approach. Within a genotype, we assemble an ensemble as before. For time alignment across genotypes, we rely on landmarks (Fig. 4a and Methods section 'Mutant time alignment').

We demonstrated this approach by analyzing how genetics affects stationary flow modules. We chose the zygotic mutant $eve^{R13}$ and the maternal double mutant $bcd^{E1}nos^{BN}$ for anterior–posterior patterning (Fig. 4b,c). Both anterior–posterior mutants were time-aligned to WT based on ventral furrow timing. As expected, $eve^{R13}$ retained DC and VF modules. In $eve^{R13}$, the germband extended less[15], but we found the GBE flow profile remained stationary (Fig. 4b). Consistent with earlier descriptions, $bcd^{E1}nos^{BN}$ exhibited DC and VF modules. We found a GBE module that was weakened compared to WT (Fig. 4c), likely due to slower overall flow[15]. We choose the zygotic mutant $twi^{ey53}$ and the maternal mutant $spz^4$ for DV patterning (Fig. 4d,e). Both DV mutants were time-aligned to WT based on cephalic furrow timing (Methods section 'Mutant time alignment'). As expected, $twi^{ey53}$ did not exhibit separate DC and VF modules, but retained a stationary GBE flow module (Fig. 4d). In $spz^4$, we found a complete loss of DC, VF and GBE modules (Fig. 4e), likely due to near complete cessation of flow.

We conclude that the anterior–posterior patterning mutants retain early WT modules, and exhibit a stationary GBE flow pattern. By contrast, intact DV patterning is required for all stationary flow patterns during early development.

### Kinematics of GBE follow a simple temperature-dependent scaling

Developmental rate is sensitive to environmental conditions such as temperature. If morphological timing is perturbed, how will tissue flows change? *Drosophila* embryos can tolerate substantial temperature variations[33]. We can thus perturb temperature to tune developmental rate without altering genotype.

We recorded live WT embryos during GBE at a series of temperatures: 17 °C ($n = 3$), 22 °C ($n = 5$), 27 °C ($n = 3$). We found that, at all temperatures, embryos were viable and the integrity of the epithelium was maintained: we did not observe noticeable differences in cell densities or apoptosis. When we measured tissue flows in these ensembles (Fig. 5a, top), we found that our temperature perturbations (1) changed flow speed monotonically and (2) did not affect the spatial pattern of tissue motion. We thus hypothesize that changing temperature simply linearly accelerates or decelerates developmental time, much like

playing an identical video at different speeds. This would be consistent with the morphogenetic program encoding target tissue deformations as 'checkpoints', rather than prescribing fixed time durations.

To test this hypothesis, we integrated tissue motion for a time duration set by relative tissue flow speed. We found that final pathlines exhibited the same patterns across temperatures (Fig. 5a, bottom). As shown in Extended Data Figs. 4 and 5, variations in tissue deformation across temperature conditions were not significantly different from variations within each condition. We found a parameter-free scaling, in which embryos achieved the same final deformations (morphodynamic milestones) at variable rates, and all velocity curves collapsed on normalizing velocity and rescaling time by the maximum speed: $v \rightarrow v/\max(v)$ and $t \rightarrow t^*\max(v)$ (Fig. 5b,c).

Does the same simple scaling govern all aspects of morphogenesis, or just tissue flows? Recent studies investigating temperature-dependent scaling have pointed to universal scaling across a survey of developmental processes[33,34], while other studies have found nonuniform scaling rules for the timing of the cell cycle and other subcellular processes[35,36]. Motivated by these variations in scaling for mitosis, we investigated this question at tissue level by measuring the onset of mitosis in distinct, spatiotemporally stereotyped 'mitotic domains' that appear on the embryo surface during GBE[37]. To measure the rate of the 'mitotic clock' at each perturbed temperature (17 °C and 27 °C), we calculated the time elapsed between the onsets of division in different mitotic domains (Fig. 5d,e and Extended Data Fig. 6). We found that the ratio of the division time differences at the two temperatures ($3.1 \pm 0.3$) diverged significantly from the ratio of maximum flow speeds that characterized the scaling of tissue motion ($1.8 \pm 0.1$) (Fig. 5d). This discrepancy was robust across different methods of measuring the time difference, such as changing reference time (Extended Data Fig. 7). Our result opens further questions: why do cell cycle and tissue deformation timings scale differently, and how are these differences accommodated to produce viable embryos over a large range of temperatures?

In contrast with GBE, the stationary GBR module did not scale with temperature (Extended Data Fig. 8). Notably, unlike GBE, GBR is not characterized by cell intercalations in the germband, but by mechanical coupling to the amnioserosa[38,39].

### Morphodynamic alignment of a shape-changing visceral organ

Morphogenesis involves tissue dynamics not only on fixed surfaces—such as the nearly ellipsoidal early embryo surface—but also on more complex tissue geometries deforming in 3D space. At a later stage of embryogenesis, tissues that invaginate during gastrulation and head involution come together to form the digestive tract[40]. During stages 15–16 of development, the midgut forms three constrictions in a sequence that divides the organ into four chambers (Fig. 6a)[41–43]. This process is known to be stereotyped across embryos, but the quantification of its dynamics has until recently proved elusive, due to the complex shapes and inherently 3D deformations[43,44]. Variability of the morphogenetic program between embryos remains largely uncharacterized[45].

During this developmental stage (unlike during GBE) the tissue at the embryo surface shows very little coherent motion, and the autocorrelation of the tissue flow on the ectodermal surface (Fig. 6b) is nearly zero. Will characterization of the complex tissue motion deep inside the embryo reveal morphodynamic modules?

To test this, we here extend our methods of spatiotemporal registration and flow correlation analysis to the development of complex shapes in the midgut. We use level sets approaches of the TubULAR package (a method developed previously in refs. 44,46) to extract a surface that penetrates ~2.5 μm within the apical surface of the endoderm, along the surface that intersects endodermal nuclei. In contrast to the static egg surface examined in the GBE analyses above, these endodermal surfaces are highly dynamic, demanding additional spatial registration steps.

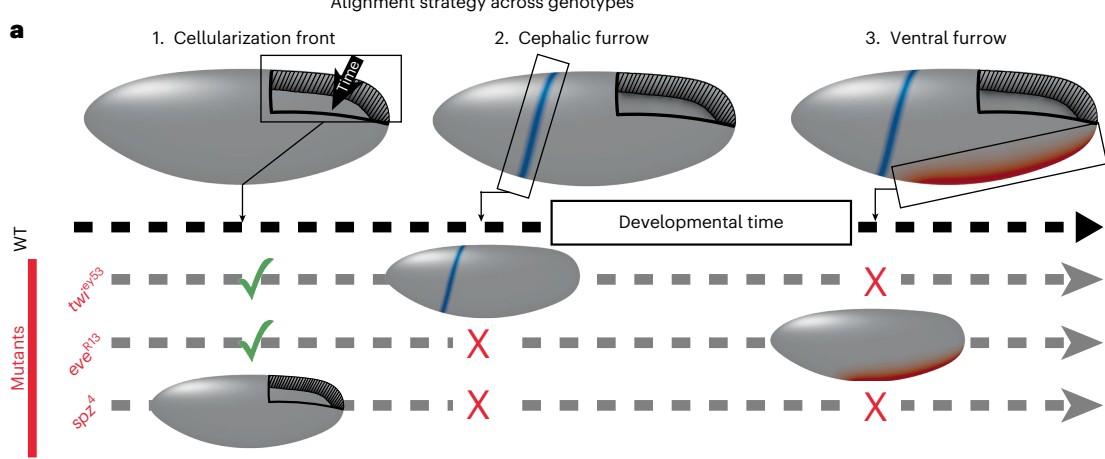

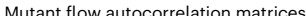

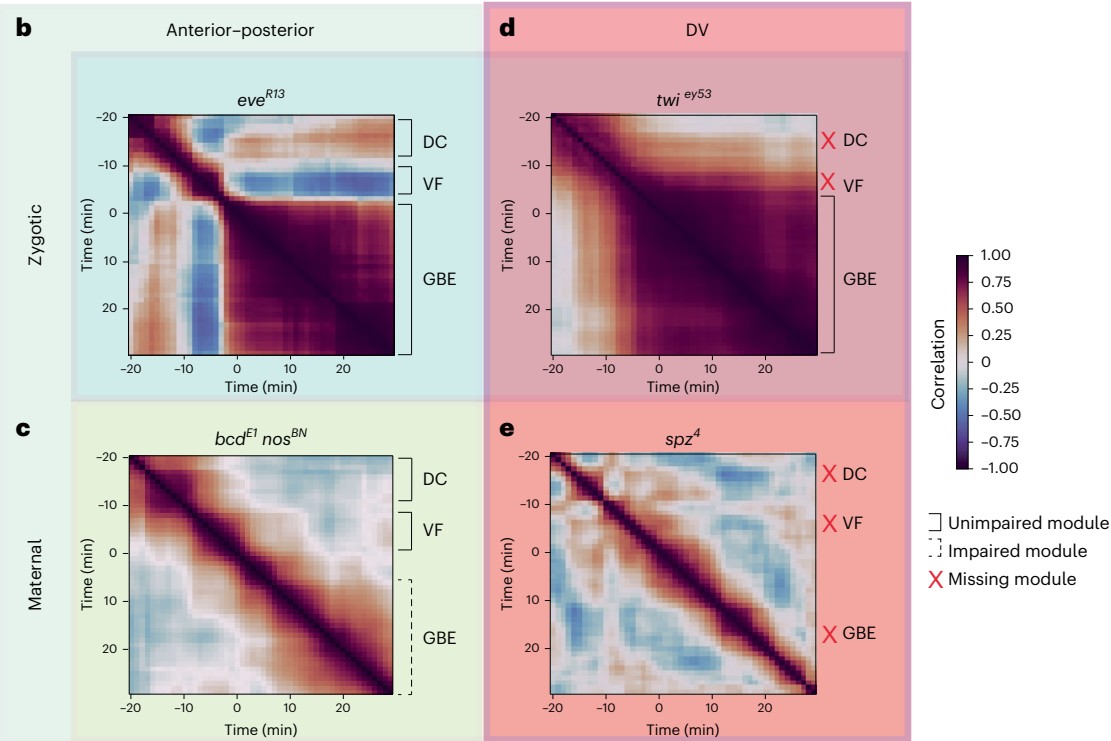

**Fig. 4 | Aligned mutant ensembles reveal DV patterning is a prerequisite for stationary WT flow modules. a**, Time alignment between mutant genotypes and WT using morphological features, shown in columns. Check marks: preserved features. Crosses: absent features. **b**–**e**, Ensemble-averaged autocorrelation of flow (vector angle method) for pattern mutant genotypes. Rigidly time-aligned such that time −15 min corresponds to WT initiation of VF (measured by first appearance of apical myosin in the furrow). Time 0 min corresponds with WT onset of GBE. Annotations: corresponding stationary WT modules from Fig. 3c. Solid black bracket: unimpaired module. Dotted bracket: impaired module (present, but distorted). Red cross: missing module. **b**, $eve^{R13}$ ($n = 3$), **c**, $bcd^{E1}nos^{BN}$ ($n = 3$), **d**, $twi^{ey53}$ ($n = 3$), **e**, $spz^4$ ($n = 3$). Average GBE module autocorrelation: **b**, 0.79, **c**, 0.57 and **d**, 0.85.

With sequences of each embryo's midgut surface in hand, we find the closest match between each pair of surfaces using iterative closest point registration (Methods section 'Spatiotemporal alignment of midgut morphogenesis'). This algorithm finds the combination of rotation and translation that best maps two 3D surfaces onto each other. This morphological approach allows alignment across embryos independent of fluorescently tagged protein (Fig. 6b(i),(ii)).

With spatial registration performed, we can quantitatively compare organ shapes to temporally align the process of midgut morphogenesis across embryos. Performing pairwise alignment across embryos leads to a consensus timeline of morphology. We demonstrated this alignment technique for an ensemble of six WT embryos, shown in Fig. 6c. We found that the rate of development through morphological stages varied by around 10%. Figure 6d shows cross-sections of these embryos in the lateral plane during four stages of development. Quantification of the shape variation across our ensemble showed the match between midgut shapes became less stereotyped as morphogenesis proceeded (Fig. 6d and Extended Data Fig. 9).

We then wondered whether the tissue kinematics exhibit periods of quasi-stationary motion, like in the early embryo's body axis elongation. We found that the autocorrelation map based on the full 3D velocity showed relatively little structure, unlike the earlier velocity autocorrelation maps of GBE (Extended Data Fig. 10). While dynamics varied continuously and thus showed some degree of correlation with

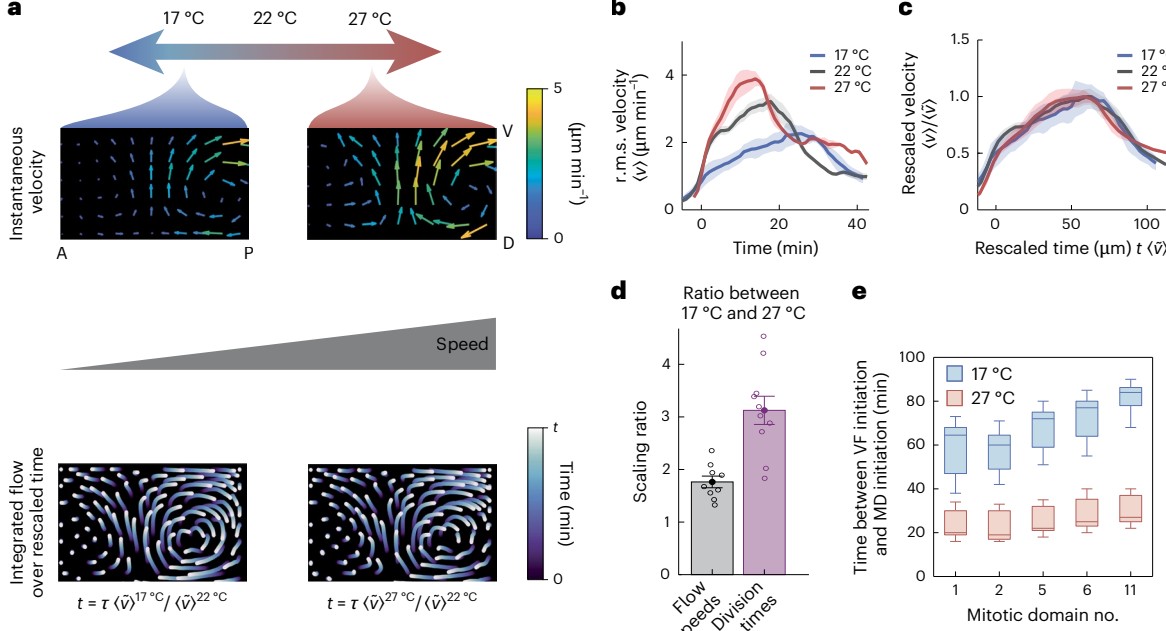

**Fig. 5 | Temperature perturbations reveal distinct scalings of GBE flows and cell divisions. a–e**, Ensembles of WT embryos were collected at three temperatures: 17 °C ($n = 3$), 22 °C ($n = 5$), 27 °C ($n = 3$). Time 0 min: onset of GBE. **a**, Kinematics of perturbed temperature conditions, with instantaneous flow (top) and integrated flow trajectories (bottom) shown for representative embryos at 17 °C and 27 °C. Displayed in 2D projections on the embryo's right half. $\tau$ denotes the GBE module duration at 22 °C, and $t$ denotes the duration of integration used, determined by $\tau$ rescaled by the maximum surface r.m.s. velocity $\langle \tilde{v} \rangle$. A, anterior; D, dorsal; P, posterior; V, ventral. **b**, Measuring the surface r.m.s. velocity $\langle v \rangle$ over time shows strong temperature dependence for embryos at 17 °C (blue), 22 °C (gray) and 27 °C (red). Shading: standard error of the mean (s.e.m.). **c**, Rescaling curves from **b** by $\langle \tilde{v} \rangle$. Speeds are rescaled by $1/\langle \tilde{v} \rangle$, and time is rescaled by $\langle \tilde{v} \rangle$. Rescaled speed is dimensionless, and rescaled time has distance units. Shading: s.e.m. **d**, Ratios of flow speeds and cell division times

between the 17 °C ($n = 3$) and 27 °C ($n = 3$) ensembles. Flow speed ratio compares speeds at ensemble maxima; datapoints show ratios between pairs of embryos at the distinct temperatures. Division time ratio compares mitotic domain initiation times, measured from ventral furrow initiation. Datapoints show ratios for left and right regions of mitotic domains {1, 2, 5, 6, 11}. Error bars show mean ± s.e.m.: flow speeds, 1.76 ± 0.11; division times, 3.12 ± 0.27. Mean ratios are significantly different: $P$ value 0.000313 (two-sided $t$-test, single comparison). **e**, The first 10–15 mitotic event timestamps were recorded in both halves of mitotic domains (MD) {1, 2, 5, 6, 11}. Data were pooled across embryos and domain, and are shown as box-and-whisker plots for distinct domains at 17 °C (blue, $n = 3$) and 27 °C (red, $n = 3$). Number of events per box plot in domains {1, 2, 5, 6, 11} respectively: 17 °C: (78, 72, 75, 74, 69); 27 °C: (86, 72, 80, 77, 73). Horizontal lines are medians, whiskers reflect full range of the data and box ranges reflect 25% and 75% percentiles.

---

similar timepoints, no distinct morphodynamic modules were present. This is intuitively sensible, as coordinate directions defined in the tissue frame of reference rotate as the surface deforms in 3D space.

We found more structure, however, in the autocorrelation of measures of tissue deformation computed in the tissue's frame of reference (Fig. 6e–g). Covariant measures of deformation—which make reference to the tissue's orientation in 3D space—did show distinct morphodynamic modules (Fig. 6g, same midgut as Extended Data Fig. 10). Moving into the reference frame of the organ to evaluate covariant measures of tissue motion highlighted boundaries between modules that were hidden in the full 3D velocity field measurement. Within each module, the autocorrelation map exhibited a box-like shape of high correlation and modules were arranged sequentially along the diagonal (Fig. 6g).

Examination of the out-of-plane deformation during these modules showed qualitatively distinct programs, illustrated for a representative midgut in Fig. 6h. During the first module (denoted as stage 15b in ref. 43), the middle constriction forms, corresponding to the purple streak bisecting the organ in Fig. 6h. Subsequently, in the second module (denoted as stage 16a in ref. 43), out-of-plane deformations appear at the anterior and posterior constriction locations. This module shares around 50% correlation with stage 15b, as the middle constriction continues to deepen. The third module (which shares substantial correlation with 16a but not with 15b) is marked by the absence of noticeable divergence or out-of-plane deformation near the posterior constriction, and a breaking of left-right symmetry in the second chamber. Finally, after constrictions complete, a new pattern

of deformation emerges during the fourth module (stage 17), in which the organ continues to coil into a helical configuration. These findings demonstrate that while the morphogenetic program is encoded in stepwise modules, not all aspects of tissue velocity will exhibit these quasi-stationary features.

More generally, our midgut analysis demonstrates that the DynamicAtlas approach—which offers a systematic strategy for discovering mechanisms of morphogenetic dynamics—can be extended to systems with complex, deforming 3D shapes.

## Discussion

Here we develop an open-source morphodynamic atlas for *Drosophila* development, DynamicAtlas, which integrates diverse classes of in toto live and fixed datasets across both space and time. The atlas includes an interactive computational platform, with both Python and MATLAB interfaces. By automatically extracting features of tissue geometry, we spatiotemporally align all datasets to a common morphological timeline. Our fast-marching-based timeline creation algorithm automatically accounts for variations in developmental rate. DynamicAtlas readily allows dynamic analysis of spatiotemporally aligned ensembles of distinct experiments.

Characterizing in toto surface tissue velocity fields in WT embryos revealed stereotyped modules of quasi-stationary flow. Temperature perturbations revealed that the duration of the GBE module scales inversely with tissue speed, such that tissues undergo a fixed total deformation, possibly encoded by the morphogenetic program.

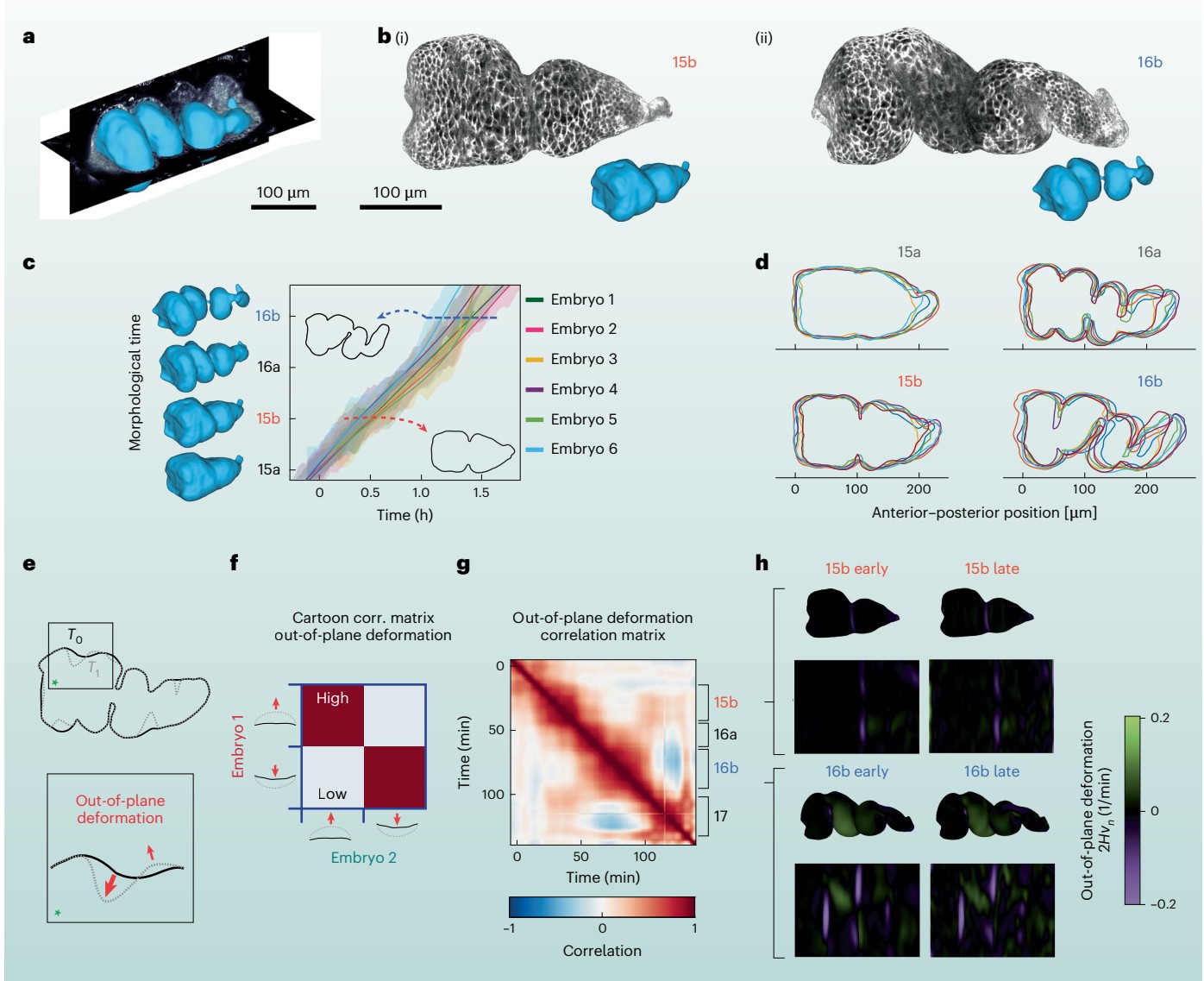

**Fig. 6 | Morphodynamic atlas of the midgut reveals modules in covariant measures of tissue deformation.** Morphological time alignment can be performed with tissue-geometry-based methods, shown in the *Drosophila* midgut. Extracting the endodermal (inner) surface of the gut using TubULAR[44] yields collections of dynamic organ shapes (rendered in blue). All midguts were imaged in WT embryos. **a**, Example extracted midgut shape in blue overlaid with orthogonal cross-sections from a 3D volumetric stack. Scale bar: 100 μm. **b**, Two endodermal tissue surfaces, labeled for a plasma membrane marker (CAAX-mCherry). Scale bar: 100 μm. (i) Stage 15b. (ii) Stage 16b. **c**, Aligning midguts onto a common morphological timeline (*n* = 6). Blue surfaces: consensus morphologies. Insets: midgut shapes from cross-sections. Shadings: 1-σ uncertainty of morphological timestamps. Time 0 h: onset of middle constriction formation. **d**, Comparison of midgut shapes across the ensemble for

the six midguts in **c**. Anterior on left. **e**, Illustration of tissue-geometry method: comparing the total out-of-plane deformation between surfaces at times $T_0$ and $T_1$. Red arrows in green-asterisk inset: deformation between surfaces in a local region. **f**, Cartoon correlation matrix of out-of-plane deformation between two shapes. **g**, Autocorrelation matrix of a representative midgut, for correlation in the tissue frame of reference (that is the Lagrangian frame). Labeled stages indicate modules. Time axes represent video acquisition time. **h**, Snapshots of the covariant out-of-plane deformation for stages 15b and 16b. Deformation heatmap shown on 3D surface representations and corresponding 2D surface projections. Deformation is measured by $2Hv_n$, where *H* is mean curvature and $v_n$ is out-of-plane motion (method from ref. 43). Inward deformation: purple. Outward deformation: green.

Mitotic events, however, scaled differently with temperature. This raises the question: are cell flows and cell divisions synchronized at subsequent morphogenetic checkpoints, for example at 'pauses' between modules? Scaling differences between morphogenetic processes have been observed in other contexts[47].

Quasi-stationary flows in the *Drosophila* embryo provide a simple physical way to establish complex shape changes. Despite the embryo's complexity, with order 10,000 cells and 10,000 genes[48], our results show the embryo executes the same flow pattern for an extended time. This stationary flow pattern is established by active processes

in cells, but does not change while cells move through it. Instead, the flow rapidly changes once the tissue reaches a target configuration.

The adherence of the GBE flow pattern to the fixed body frame is reminiscent of the spatial precision of blastoderm gene expression patterns with respect to the body axes[49,50]. This presents a challenge: given that the flow pattern remains aligned with the egg's principal axes while cell positions (and hence patterns of nuclear gene expression) continuously change, how does genetic signaling in cells instruct flow directions? The global myosin distribution predicts tissue flow during GBE[22]. The stationary GBE flow pattern thus suggests that the myosin

pattern is also stationary in the body frame. This is consistent with recent studies showing that static DV-aligned cues control the global myosin pattern during GBE[28], and that cell–cell interfaces replenish their myosin level according to their degree of DV-axis alignment[51]. Here we found that stationary flow patterns require DV, not anterior–posterior, patterning. Taken together, this adds to growing evidence that the DV patterning system coordinates robust morphogenetic movements of GBE, for example by patterning mechanical feedback[52,53].

DynamicAtlas enables broad applications. Atlas data are directly amenable to force inference methods[54] or to machine-learning algorithms[55]. In addition, similar to other studies[14], our atlas could be integrated with scRNA-seq data, allowing exploration of the interplay between mechanics and cell fate, for example by comparing cell trajectories in real space to their trajectories in transcriptomic space[11].

Our atlas has important limitations in both data and software. Most atlas data were recorded either during gastrulation or midgut development, with limited data outside those stages. Furthermore, RNA transcripts were not labeled, limiting exploration of dynamic transcriptional control during tissue motion. Software-wise, our atlas does not contain methods for the potential broader analyses noted above, and our computational tools are structured around analyzing surface projections, with limited methods for exploring interior volumes. However, we emphasize that DynamicAtlas is designed to allow smooth incorporation of future datasets and analysis tools.

We anticipate that DynamicAtlas will provide a versatile data repository and analysis platform for the wider *Drosophila* community. More generally, our framework enables in toto spatiotemporal alignment of embryos using reproducible features of developmental dynamics. We hope this work may serve as a template for quantitative, dynamic analysis of morphogenesis in other living systems.

## Online content

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

## Methods

### Ethics statement

All experiments were conducted in the USA, in accordance with the Animal Welfare Act, Health Research Extension Act, and National Institutes of Health regulations. All experiments were performed with invertebrate organisms (*D. melanogaster*): thus, ethics oversight was not required.

### Statistics and reproducibility

No statistical method was used to predetermine sample size. No data were excluded from the analyses. The experiments were not randomized. The Investigators were not blinded to allocation during experiments and outcome assessment.

### Time alignment for early embryo datasets

As explained in the main text, to align two live recordings of embryo development (denoted A and B), we need (1) a method to quantitatively score the similarity between a frame from A and a frame from B, and (2) a method to convert those pairwise scores into a single, consistent mapping of the timepoints of recording A to those of recording B. Consistency requires, in particular, that the mapping is order preserving (mathematically speaking, monotonously nondecreasing). Here we explain (1) the quantitative scores and then (2) the timeline-construction algorithms. Finally, we describe how to obtain a single consensus timeline from multiple (more than two) videos, and how we timestamp fixed data. For more details on the following subsections, see Supplementary Note 6.

**Aligning live Runt nanobody datasets.** To measure the similarity of datasets containing a Runt marker (either a Runt nanobody in the case of live embryos or a Runt staining in the case of fixed embryos), we computationally extract the anterior boundary of Runt stripe 7 using a custom image processing script. These curves, as shown in Fig. 2, can be compared quantitatively to obtain a similarity score. This measure is independent of image details such as signal to noise ratio, contrast or visibility of individual nuclei, allowing a comparison between fixed and live data. The form of Runt stripes has been shown to track the overall tissue deformation during axis elongation[28].

**Aligning live datasets using tissue deformation.** To align early embryo live data without PRG expression, we use an alternative measure of tissue deformation. As explained above, using PIV we can compute tissue flow fields in a robust and automated way. However, the velocity field itself is not suitable for timestamping, since it does not change much during axis elongation (Fig. 3). We therefore integrate the velocity field (rates of tissue movement) to obtain the amount of total tissue deformation. This integration was carried out numerically using the MATLAB implementation of the Runge–Kutta method. To define the ordinary differential equation system for the Runge–Kutta method, we linearly interpolate the velocity field, defined on the PIV grid, in both space and time. Total tissue deformation (in the form of a vector indicating total displacement at each point on a grid over the embryo surface) can then readily be compared across embryos for time alignment.

However, to measure deformation, we need to decide which timepoint in each video we consider as the un-deformed reference state and use as the starting point $t_0^\omega$ of integration. To make a choice of initial time that is the same across different embryos, we use the fact that the velocity field changes very rapidly at the onset of axis elongation. This timepoint of rapid change can be identified from the velocity field alone. More information about how this is done is given in Supplementary Note 6.

**Fast-marching algorithm.** We now describe how we use the frame-to-frame pairwise comparison metrics shown above to match two timelines. While we use rigid time alignment ('Rigid time alignment' section) for analyses of the reproducibility and quasi-stationarity of the flow in the main text, the atlas data are timed using a time warping procedure built on fast marching. The fast-marching-based method accommodates variable rates of progress along the morphogenetic program, as described in the main text.

Given two videos A and B, we denote the time variable of the videos by $t_A$ and $t_B$, respectively. The fast-marching algorithm outputs a correspondence curve $s$ that maps $t_A \mapsto t_B$. In case videos A and B do not cover the same period of developmental time (for example, one video starts earlier), the correspondence curve is only defined for the period of overlap. This curve is guaranteed to be monotonously increasing, and therefore invertible where defined. However, the correspondence curve need not be a straight line with unit slope, which is what makes it possible to accommodate variable developmental rates, as well as 'fits and starts' in developmental progress.

The correspondence curve is computed using the following procedure:

(1) Using a suitable comparison metric (for example, the Runt stripe-7 comparison explained above), calculate the similarity matrix: for each pair of times $(t_A, t_B)$, compute the similarity of the two frames from videos A and B. This creates a matrix $M$ where rows correspond to timepoints in video A, and columns to timepoints in video B (Extended Data Fig. 1a).

(2) Find the startpoint and endpoint of the correspondence curve. These two points lie on the edges of the similarity matrix (top or left edge for the startpoint, bottom or right edge for the endpoint). Let us assume that video A starts earlier than video B. In this case, the startpoint of the correspondence curve is in the first column of the matrix (left edge), at the row where the similarity measure is maximal (white point in Extended Data Fig. 1a). The endpoint is similarly chosen as the point on the right (if video A ends later than video B) edge with maximal similarity measure. The DynamicAtlas software shows an interactive prompt so that the user can check, and if necessary, modify, the chosen startpoint and endpoint.

(3) Find the correspondence curve. Using a fast-marching algorithm, we find the weighted shortest path from startpoint to endpoint through the similarity matrix. The weighted length of a curve segment is the geometric length of the curve segment, times the inverse of the similarity metric along the segment. This means that the 'cost' for the correspondence curve to traverse a region where frames are very dissimilar is high. If the similarity metric is constant, the shortest weighted path is a straight line. When the similarity metric is not constant, the shortest path will bend to move along the ridge of maximal similarity, as shown in Extended Data Fig. 1a. The fast-marching algorithm is restricted to consider paths that always move downward or rightward through the matrix, which ensures that the correspondence curve is order preserving.

**Rigid time alignment.** The fast-marching algorithm allows for nonstraight correspondence curves between the timelines of different live datasets, which accommodates morphogenetic variability. For many purposes, however, we want to process and distort the data as little as possible. For example, in Fig. 3, we want to measure the amount of variability between the flow fields of different embryos. For Fig. 3e(i),(ii), we therefore perform only the simplest time alignment: a rigid time alignment that shifts each timeline by a fixed amount. To do this, we choose a landmark event and then shift all videos so that the landmark event occurs at the same time.

We now detail how we define this landmark event, corresponding to early GBE. We make use of the autocorrelation matrices of the instantaneous tangent velocity field, shown in Fig. 3a. These are defined in the

correlation section below (Methods section 'Correlations of velocity fields') as well as in Supplementary Note 10.

For every individual embryo $e$ in the ensemble recorded for a discrete $T$ frames, with instantaneous surface tangent velocity fields $\mathbf{v}(t)$, a square autocorrelation matrix $M_e$ of side length $T$ was computed, such that $M_e(t_i, t_j) = \rho(\mathbf{v}(t_i), \mathbf{v}(t_j))$. For purposes of alignment, we defined $\rho$ by the vorticity correlation measure (Supplementary Note 10). We chose this definition because at the transition from VF to GBE, the vorticity pattern in the posterior exhibits a clear sign change and $\rho_\omega$ robustly captures this phenomenon (VF strongly correlates with itself, but strongly anti-correlates with GBE).

All videos in the ensemble used for Fig. 3 included this transition, which provided a natural time to align the videos for analysis of the quasi-stationary flow. Interpreting the matrix in Cartesian coordinate space $(x, y)$, the alignment time $t_0^e$ for the embryo was computed by:

$$t_0^e = \arg\max_x \left( \sum_y \partial_x M_e(x, y) \right)$$

All embryos were then aligned by rigidly shifting their time coordinates such that all $t_0^e$ were equivalent to a common $t_0^\omega$, before any temporal comparisons of flow were performed (for example, temporal averaging; Supplementary Note 11).

**All-to-all alignment-and-consensus algorithm.** In the case of rigid alignment, there is no real difference between aligning two embryos or more than two embryos, since the time alignment is a simple shift based on a landmark event. This is not true for fast-marching-based alignment where the correspondence curves are not straight lines with unit slope anymore.

One simple way out is to designate a single recording of good quality as reference, and use pairwise alignment to match all other recordings to the reference. However, this is not robust because it involves arbitrary choice, and the resulting timeline is bound to the potential peculiarities of the reference recording. For details on the alignment algorithm used to overcome this problem, see Supplementary Note 6.

**Spatiotemporal alignment of midgut morphogenesis.** To align midgut shapes to one another in space and time, we scan for pairwise correspondences in separate embryos' timelines, as before. We compare each timepoint of a given midgut shape sequence to an optimally oriented shape of another shape sequence. For details, see Supplementary Note 7.

### Stocks and antibodies used in the atlas
In the Supplementary Information, lists of *Drosophila* stocks (Supplementary Table 1 and Supplementary Note 2) and primary antibodies (Supplementary Table 2) are available. These tables also enumerate which resources were used to generate main text figure panels as well as supplementary videos. Stocks and reagents for the midgut data contained in the DynamicAtlas are described in detail in ref. 43.

### Immunohistochemistry
Fixed *Drosophila* embryos in the atlas were collected in batches at 4 h postfertilization and dechorionated with 50% bleach (Chlorox). For heat and/or methanol fixation, embryos were fixed according to standard protocols[58]. For paraformaldehyde (PFA) (Electron Microscopy Sciences) fixation, embryos were incubated on a rocker for 25 min in a solution of 5 ml of heptane and 5 ml of 4% PFA diluted in 1× PBS. After fixation the aqueous fixative layer was removed and embryos were devitellinized by adding 5 ml of methanol and vortexing at maximum speed for 1 min. Devitellinized embryos were collected and rinsed 3 times with methanol and stored at −20 °C. Primary antibodies and dilutions are described in Supplementary Table 2. Donkey and goat secondary antibodies conjugated to Alexa Fluor 488, 568 and 647 were used (1:500, ThermoFisher Scientific).

### Microscopy
For all datasets, we used a custom Multi View Selective Plane Illumination Microscope (MuVi SPIM)[17] with scatter reduction through confocal imaging[16]. This enabled fluorescent imaging of the entire *D. melanogaster* embryo at subcellular resolution, as in references[43,52]. Embryos were dechorionated according to standard procedures and mounted in 1.5% low-melting point agarose (Millipore Sigma-Aldrich) for imaging in the MuVi SPIM. Electronics were controlled using MicroManager[59]. Since the custom MuVi SPIM's sample chamber is water filled, it can be equipped with temperature control, which keeps the water at a fixed temperature during imaging. This method was used to generate the datasets at 17 °C and 27 °C. Details of deep tissue imaging for midgut datasets are described in detail in ref. 43. For fusion and deconvolution of the light-sheet data, open-source software package ImageJ[60] was used with the Multiview Deconvolution plugin[18]. Certain visibly obvious imaging and fusion artifacts (Supplementary Note 3) specific to in toto light-sheet microscopy can occur infrequently using the data collection methodologies we employed. To account for these rare effects (which visibly and obviously manifest during the multiview fusion and cartographic surface projection phases of data processing), datasets that did not successfully fuse and/or datasets in which the surface projection was not optimal were excluded from inclusion in the atlas.

### Tissue cartography
We use the fact that during gastrulation, much of the embryo tissue is nearly confined to a simple quasi-2D ellipsoidal shape, to computationally map each embryo-timepoint to a fixed 2D domain using ImSAnE[23], enhanced with surface detection features using level sets[44,46]. During gut development, the gut tissue similarly forms a 2D surface: however, this surface is dynamic. Technical details of these methods are available in refs. 23 and 44. For more details on the computational steps, see Supplementary Notes 8 and 9.

### Quantification of tissue deformation with PIV
We used PIV to compute the instantaneous tangent velocity field of the embryo surface tissue for all the videos in our ensemble. The PIV algorithm divides an image into smaller windows and measures the displacement of each window between subsequent timepoints. Therefore, PIV yields a measure of tissue velocity on a coarser grid, whose spacing equals the window size. We compute the PIV on the vertices of a square lattice in coordinate space $(x_1, x_2)$ with edge length of ~2–3% of the range in both directions (hence, spanning multiple cell widths). Here $x_1$ parametrizes the longitudinal coordinate along the embryo's length and $x_2$ parametrizes the circumferential coordinate along the DV axis. To avoid edge effect at the cut in the periodic dimension $x_2$ of the cylindrical chart, we computed PIV on images tiled in $x_2$. To account for distortions at the poles that worsened PIV detection ($x_1$ close to its minimum or maximum), for all subsequent computations we truncated ~8% of the $x_1$ coordinate at both ends of its range. PIV fields were computed using a custom MATLAB script, and the PIVlab MATLAB plugin[30] when better spatial resolution was required.

This procedure was followed for every embryo in the ensemble, resulting in a spatially discrete velocity field $\mathbf{v}_{embryo}(t, x_1, x_2)$ with a vector at each point of the lattice grid. Since all 2D map projections were generated in the same coordinate space, all resultant velocity fields were directly comparable at corresponding coordinate locations between embryos. Finally, using the known embedding functions $\mathbf{r}(x_1, x_2)$ (that is, the 3D positions corresponding to each point on the PIV grid) and the implied metric tensor, the velocity fields were converted into real 3D space, resulting in a physical spatiotemporal tangent velocity field $\mathbf{v}_{embryo}(t, \mathbf{r})$ on the surface of each embryo. Here $\mathbf{r}$ denotes a position in 3D embedding space (the laboratory reference frame).

### Correlations of velocity fields
Several parts of our analysis required quantitative comparison of two velocity fields $\mathbf{v}_{e_i}(t)$ and $\mathbf{v}_{e_j}(t')$ between two embryos $e_i$ and $e_j$ at times

$t, t'$ (including the autocorrelation case where $i = j$). We defined a matrix $S$ whose scalar entries $S_{tt'}$ are given by a 'similarity' measure $\rho(\mathbf{v}_{e_i}(t), \mathbf{v}_{e_j}(t'))$ that defines correlation between the velocity fields at the respective times.

We made use of several similarity measures $\rho$ between two velocity fields. In the case of early embryo data, these vector fields are defined over the same spatial coordinates—namely, those of the embryo surface—and can be readily computed using the single parametrization for the early embryo surface. In Extended Data Fig. 2, we show that the result of the main text holds irrespective of which similarity measure we use to evaluate correlations between velocity fields. Details on these measures are described in the Supplementary Information (Supplementary Note 10).

### Statistical comparison of embryos at varying temperatures
We reported in the main text that the tissue deformations during GBE in embryos at low temperatures match the tissue deformations in embryos at high temperatures once time is rescaled by the maximum speed. This led to a single, parameter-free scaling of morphogenetic tissue deformation during GBE. Here we supply additional detail.

To time align the different datasets during GBE, we use landmark-based rigid alignment. For all samples, we first measure speed averaged across the embryo (Supplementary Note 13). This gives curves of speed as a function of time $|v(t)|$, which we align to each other by looking for the timepoint of maximum acceleration, that is maximum of the derivative of $|v(t)|$. This defines our reference time $t = 0$ for all temperature conditions. We also use this method to compute the time we denote as the 'onset of GBE' in the text.

To compare tissue deformations between temperature conditions, we first define the tissue displacement by integrating the coarse-grained tissue velocity fields over time using a Runge–Kutta fourth order scheme. For the purposes of numerical integration we linearly interpolate the velocity field (known at grid points) in space and time. We chose a numerical integration timestep of 0.2 min for smoothing the trajectories (the value was chosen such that multiple smoothing timesteps are integrated within our time resolution of 1 min, but its value is arbitrary). For a piece of tissue that begins at $(x_0, y_0)$ at time $t = 0$ and moves, according to PIV integration, to $(x_1, y_1)$ at a later time $t_1$, we then compute the displacement field as

$$\Delta(t_1) \equiv \sqrt{(x_1 - x_0)^2 + (y_1 - y_0)^2}$$

We can compare the displacement fields across different temperature conditions to see whether the time rescaling proposed in the main text can completely account for the differences in tissue deformation between temperature conditions. To do this, we choose the endpoint of PIV integration $t_1$ in a temperature-dependent way: the endpoint is taken at $t_1(T) = t_1(22\,°C) \times \frac{v_{\max, 22\,°C}}{v_{\max, T}}$. The second factor is the maximum-velocity-based rescaling factor introduced in the main text. This means we integrate for a longer time for cold embryos, and for a shorter time for hot embryos. This defines the displacement fields $\Delta_T$ for the three temperature conditions.

For each condition, we have imaged at least three embryos ($n = 3$ for 17 °C, $n = 5$ for 22 °C, $n = 3$ for 27 °C). We can therefore compute the mean $\overline{\Delta_T}$ and variance $\sigma_T^2$ of $\Delta_T$ over samples $i$ within each temperature condition. This then allows us to test statistically whether the difference in the means $\overline{\Delta_T}$ of different conditions is significant, compared to in-condition variance. We find that the displacement fields of different conditions do not differ by more than expected from in-condition variance, confirming that the simple rescaling relation accounts completely for the change in tissue deformation by temperature.

For the analysis of GBR in Extended Data Fig. 8, we recorded one WT video encompassing GBR at each temperature condition of 21 °C, 23 °C, 25 °C and 27 °C. We rigidly aligned these videos to the time when their root-mean-square (r.m.s.) velocity was maximal, and just as during GBE, we defined time 0 in their common timeline as the onset of GBE

($\Delta t$ between the two times was determined using the extended dataset in Fig. 3b). Standard deviations (s.d.) were computed over space. The anterior half of the embryo exhibited low flow and was truncated during computations of mean velocity and s.d.

### Timing of mitotic domains
We timed the appearance of mitotic events using live sqh:GFP datasets taken at 17 °C ($n = 3$) and 27 °C ($n = 3$). Mitotic events display a sudden appearance of a bright myosin band across a cell that typically resolves into 2 cells in the subsequent frame 1 min later, as shown in Extended Data Fig. 6a. We therefore recorded the onset of the first 10–15 events in each domain by manual inspection, marking the timepoints at which mitoses appear in each domain. Extended Data Fig. 6b shows the locations of the mitotic domains tracked on the embryo surface.

In Extended Data Fig. 7, we detail how we compute the scaling factor describing the acceleration or deceleration of the mitotic clock at different temperatures. We measured the ratio of mitotic speed in 17 °C and 27 °C in 2 different ways: first, for each of the 5 mitotic domains considered, we computed the time elapsed between the acceleration-based $t = 0$ defined above and the first division, $\Delta t^{27\,°C}$ and $\Delta t^{17\,°C}$. We also calculated the time elapsed since the time of vorticity sign reversal $t_0^\omega$, also defined above. Second, we paired the mitotic domains up (for example, into the pair 'domain 5--domain 11') and calculated the time $\delta t$ between the onset of division in the first and the second domain in the pair.

### Correlations of images
For computing correlations between images, used in computing similarity between PRG datasets (including Runt stripes), the Pearson correlation coefficient was always used. Given two images (that is, matrices) $A$ and $B$, and given their respective mean intensities $\bar{A}$ and $\bar{B}$, their Pearson correlation is given by:

$$r = \frac{\sum_m \sum_n (A_{mn} - \bar{A})(B_{mn} - \bar{B})}{\sqrt{\left(\sum_m \sum_n (A_{mn} - \bar{A})^2\right)\left(\sum_m \sum_n (B_{mn} - \bar{B})^2\right)}}$$

This correlation measure returns a value from −1 (perfectly anticorrelated) to +1 (perfectly correlated), and is robust to varying mean intensity levels across images.

### Correlation of midgut deformations
In midgut datasets, out-of-plane deformations were computed and used to calculate tissue-frame (Lagrangian) correlations. Out-of-plane deformations were measured and mapped to 2D surface projections using the methods in ref. 43. Correlation between projections was then computed as in the 'Correlations of images' section above.

### PRG decorrelation
Two methods were used to determine the approximate timescale of decorrelation of PRG expression patterns during GBE: an empirical observation $t_{obs}$, and an estimation $t_{est}$. At 1 s.d., a Gaussian function decreases to a factor of $e^{-\frac{1}{2}} = \frac{1}{\sqrt{e}} \approx 0.61$ of the peak, and we defined two PRG patterns as 'decorrelated' when their correlation fell below this threshold.

For the observation $t_{obs}$, a Gaussian fit was performed on the curve in Fig. 3h, using a two-term sum of Gaussians function. The s.d.s of the 2 Gaussians differed by a factor of 3.8, and the smaller s.d. was used to estimate the decorrelation timescale. The narrower Gaussian s.d. was $t_{obs} = 2.2$ min, with 95% confidence interval $t_{obs} \in (1.4, 3.0)$ min.

For the estimation $t_{est}$, characteristic length scale was computed using Runt stripe widths. These were estimated for 6 Runt samples at aligned time of 46 min following the onset of GBE, corresponding with the Gaussian peak used for $t_{obs}$. Stripe widths were estimated as twice the s.d. of the intensity curve computed along a line manually drawn between stripe boundaries, at a DV coordinate in the neuroectoderm.

Widths for Runt stripes 4, 5 and 6 were measured and averaged within each sample. Across the 6 samples, this yielded an estimated stripe width of $w_{est} = 19.0 \pm 1.7$ μm (mean ± s.d.). The characteristic speed scale at this time was estimated from the r.m.s. velocity curve in Fig. 3i, yielding $v_{est} = 3.13 \pm 0.20$ μm min$^{-1}$. Using the nonnegative periodic function $f(x) = \cos^2(x)$ as a heuristic for stripe pattern, we found that when two patterns of spatial wavelength $\lambda$ were shifted by a distance $\Delta x$ such that the dimensionless quantity $c \equiv \frac{\Delta x}{\lambda} \approx 0.34$, the time $t_{est} = \frac{c \times w_{est}}{v_{est}}$ yielded the above decorrelation threshold. This gave $t_{est} = 2.1 \pm 0.2$ min (mean ± s.d.).

### Uncertainty of morphological timestamping
For timestamping samples onto a morphological timeline (as depicted in Figs. 2c(iii) and 6c), the uncertainty $\sigma_t$ of a timestamp $t_0$ was reported as a 1-$\sigma$ confidence interval: $(t_0 - \sigma_t, t_0 + \sigma_t)$. Given chi-squared as a function of time $\chi^2(t)$, derived from a morphological measure of choice (for example, the comparison of a geometric feature between sample and timeline), the value for $\sigma_t$ is related to the curvature of the chi-squared value at $t_0$:

$$\sigma_t^2 = 2\left(\frac{\partial^2 \chi^2}{\partial t^2}|_{t=t_0}\right)^{-1}$$

as documented in ref. 61. To determine this empirically for each $t_0$, a parabola was fit to $\chi^2(t)$ at $t_0$ and $\sigma_t$ was then found from the curvature of the fitted parabola.

The uncertainty of timestamping onto a particular timeline depends on the reproducibility of the morphological feature used for alignment, and the number of samples (minimum $n = 3$ required). For the timelines in our analysis, we note that our time alignment is based on live videos of a highly reproducible process. Each dataset in our $n = 3$ ensembles (for example, our 17 °C and 27 °C ensembles in Fig. 5) contains on the order of 50 to 100 datapoints, one per frame. Our alignment method uses all timepoints in all embryos and therefore uses at least an order of magnitude more datapoints than the number of samples. Because of this quantity of information, we can use the live alignment to create a robust morphological timeline, which can then incorporate many more samples, including fixed datasets (Fig. 1e).

### Mutant time alignment
For the mutant data analyzed in the main text, four mutant genotypes were examined: $eve^{R13}$ ($n = 3$), $twi^{ey53}$ ($n = 3$), $bcd^{E1}nos^{BN}$ ($n = 3$), $spz^4$ ($n = 3$). Autocorrelation maps shown in Fig. 4b–e were generated for these data after time-aligning the ensembles within genotype and across genotype, using the methods below and in Supplementary Note 12.

Within each genotype, ensemble alignment was performed by computing the rigid time alignments between every pair of samples, such that the correlations of the respective PIV velocity curves were maximized. Adjusting for these temporal offsets then aligned the overall ensemble for the genotype.

Across genotypes, alignment was performed by synchronizing discrete morphological events of each genotype to the WT timeline, using the strategy depicted Fig. 4a. In specific, the anterior–posterior patterning mutants $eve^{R13}$ and $bcd^{E1}nos^{BN}$ were aligned to WT using the onset of VF, while the DV patterning mutants $twi^{ey53}$ and $spz^4$ were aligned to WT using the onset of cephalic furrow formation. Both onsets were defined by the first appearance of apical myosin in the future furrow. After these alignments were performed, all ensembles then corresponded to WT ventral furrow onset at the same time coordinate (time −15 min).

### Reporting summary
Further information on research design is available in the Nature Portfolio Reporting Summary linked to this article.

### Data availability
Experimental data contained within the atlas (including the live datasets used to perform analysis of GBE) are publicly available via Dryad at https://doi.org/10.25349/D9WW43 (ref. 19). This includes the spreadsheet of metadata used by the Python interface to query data (MorphodynamicAtlas.csv). We have also included a minimal 'demo dataset' (DEMO_DATASET.tar.lz4), used in the MATLAB tutorial contained in the Supplementary Information (MATLAB Interface Tutorial). This demo dataset is available via Zenodo at https://doi.org/10.5281/zenodo.14792464 (ref. 62). Source data are provided with this paper.

### Code availability
DynamicAtlas code for the Python-based interface is available via Zenodo at https://doi.org/10.5281/zenodo.14285126 (ref. 20). DynamicAtlas code for the MATLAB-based interface is available via GitHub at https://github.com/npmitchell/dynamicatlas (ref. 21). To aid new users, the end of the Supplementary Information includes tutorials that demonstrate the core functionality of the Python and MATLAB interfaces on demo datasets (Supplementary Information Python interface tutorial and MATLAB interface tutorial). The MATLAB demo script is reproduced in Supplementary Note 1.

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

### Acknowledgements
We thank E. Wieschaus, F. Brauns, D. Cislo and T. Bibilashvili for insightful discussions. We thank all current and former members of the Streichan Laboratory for critical feedback, helpful support and fostering a stimulating scientific environment. E. Wieschaus contributed stocks and reagents. S.J.S. acknowledges funding by National Institutes of Health grant no. R35 GM138203 and National Science Foundation grant no. PHY-2047140. A.R.B. acknowledges funding from the European Research Council under the European Union's Horizon 2020 research and innovation program (grant agreement no. 810104-PoInt). The content is solely the responsibility of the authors and does not necessarily represent the official views of the National Institutes of Health or the National Science Foundation.

### Author contributions
M.F.L., N.P.M., M.K.R., H.J.G., F.E.S. and S.J.S. performed data collection. M.F.L., V.J.-S., N.P.M., M.K.R., H.J.G. and S.J.S. performed data postprocessing. V.J.-S., N.C. and N.P.M. analyzed the data. N.P.M. and V.J.-S. developed the DynamicAtlas MATLAB interface. N.C. developed the DynamicAtlas Python interface. M.F.L., N.P.M. and S.J.S. wrote the original draft of the paper. V.J.-S., M.F.L. and S.J.S. reviewed and edited the final paper. A.R.B. and S.J.S. acquired funding. S.J.S. supervised the project.

### Competing interests
The authors declare no competing interests.

### Additional information
**Extended data** is available for this paper at https://doi.org/10.1038/s41592-025-02897-8.

**Correspondence and requests for materials** should be addressed to Sebastian J. Streichan.

Peer reviewer reports are available. Primary Handling Editor: Madhura Mukhopadhyay, in collaboration with the *Nature Methods* team.

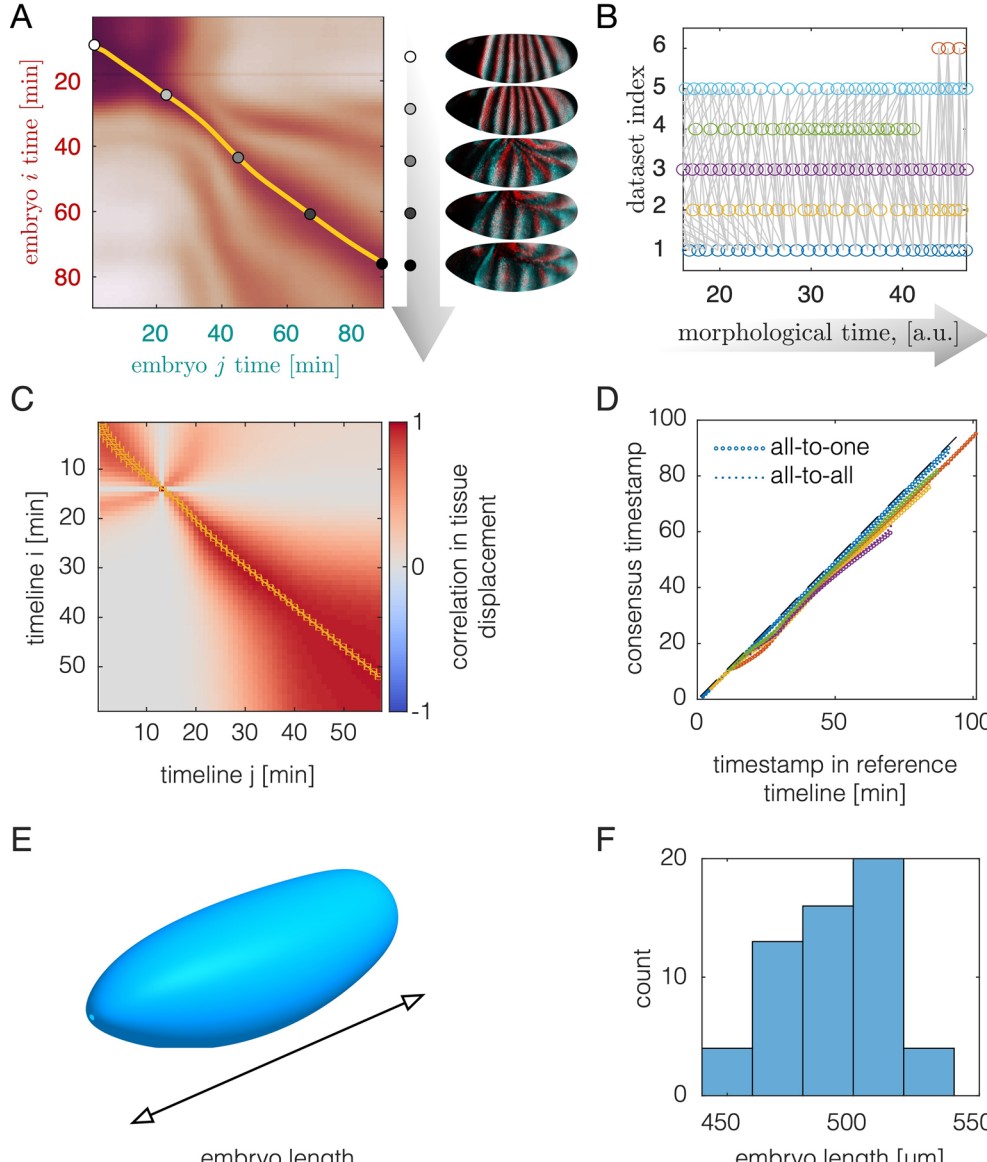

**Extended Data Fig. 1 | Creation of a morphological timeline.** *(A)* For two Runt nanobody embryos (same embryos as Fig. 2b(ii)), we compute the similarity of each pair of frames, yielding a similarity matrix. Timeline algorithm identifies a startpoint (white) and endpoint (black) and routes a correspondence curve (orange) along the ridge of maximal similarity. *(B)* Using a wire-network inspired algorithm, we use pairwise time alignment to create a consensus timeline for multiple samples. *(C)* Tissue deformation-based similarity matrix computed from PIV for two sqh:GFP embryos. The integration reference point is visible as the 'cross' of zero correlation in the heatmap. Fast marching through the matrix returns a correspondence curve (yellow points) with uncertainties estimated by the variance between the raw maxima in each row and the computed fast marching curves. *(D)* Comparison of all-to-all and all-to-to-one (reference-recording based) timeline construction for five sqh:GFP datasets. Both methods yield similar results. *(E)* To characterize possible mapping error, we measured the AP-axis length of the embryo, from anterior pole to posterior pole. *(F)* Histogram of AP-axis lengths for 57 wild-type embryos from the atlas (same embryos as in Fig. 3e(ii)). Mean ± SD: 492 ± 20 $\mu$m. SEM: 3 $\mu$m.

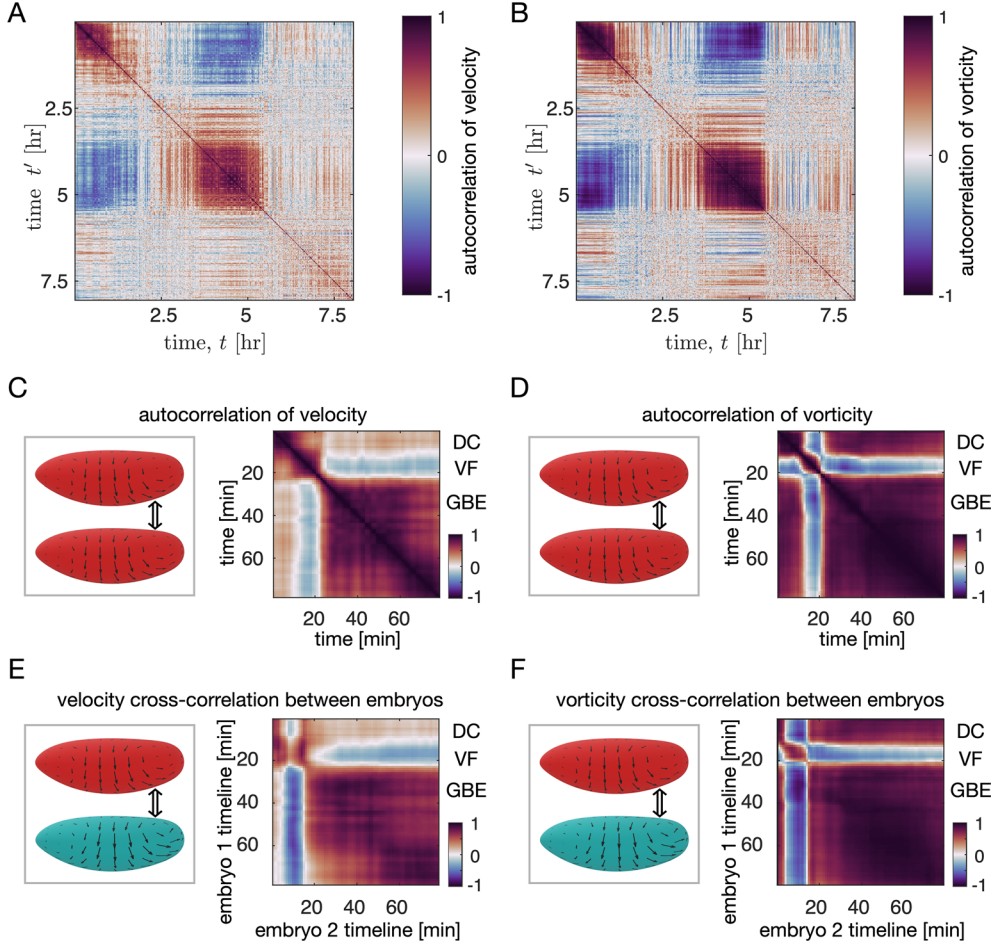

**Extended Data Fig. 2 | Autocorrelation of surface flow yields similar stationary modules irrespective of correlation measure.** *(A)-(F)* Time axes represent movie acquisition times. Wild-type embryos. (A) Measurement of the velocity autocorrelation $\rho^v$, defined in SI Eqn. 6 as the normalized inner product of velocity vectors on the surface, shows blocks of quasi-stationary flow. Live embryo dataset recorded from onset of GBE to dorsal closure. *(B)* Measurement of the vorticity autocorrelation $\rho^\omega$, defined in SI Eqn. 8, shows similar blocks of quasi-stationary flow. Same embryo and timeline as (A). *(C)* A representative embryo's velocity-based autocorrelation during early development shows the DC, VF and GBE modules, described in Fig. 3. *(D)* Vorticity-based autocorrelation of the same embryo as (C) also shows the DC, VF, and GBE modules. *(E)* Velocity-based cross-correlation of two live embryos during early development shows DC, VF, and GBE modules are maintained between the pair. *(F)* Vorticity-based cross-correlation of the same pair of embryos as (E) also shows the same DC, VF, and GBE modules between the two.

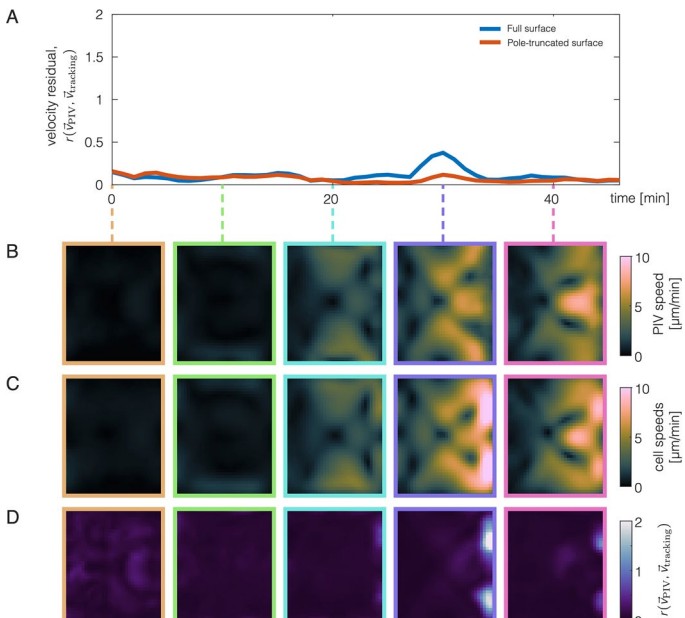

**Extended Data Fig. 3 | Velocity fields derived from PIV match velocity fields derived from cell tracking ('PTV'), except near the posterior pole.** *(A)-(D)* As a control for PIV, we track cells in a live, membrane marker-labeled dataset, allowing the velocity field to be computed from both PIV and PTV in the same embryo (CAAX-mCherry, n=1, shown in Supplementary Videos 2 and 3). Time axis represents movie acquisition time. *(A)* The spatially-averaged velocity residual (computed using SI Eqn. 11) between PIV and PTV remains small throughout development. The blue curve is the average residual across the full embryo surface. The red curve is the average residual across the middle 70% of the embryo surface. This ensures that regions near the poles, where distortions on our 2D map projections are high (at the anterior-most 15% and posterior-most 15% of the surface), are truncated in the comparison. *(B)-(D)* Comparison of PIV and PTV at different points in developmental time shows good agreement. *(B)* Magnitude of PIV (PIV-derived speed) on the embryo surface. *(C)* Magnitude of PTV (tracking-derived speed) on the embryo surface. *(D)* Residual between PIV and PTV on the embryo surface, computed using SI Eqn. 10.

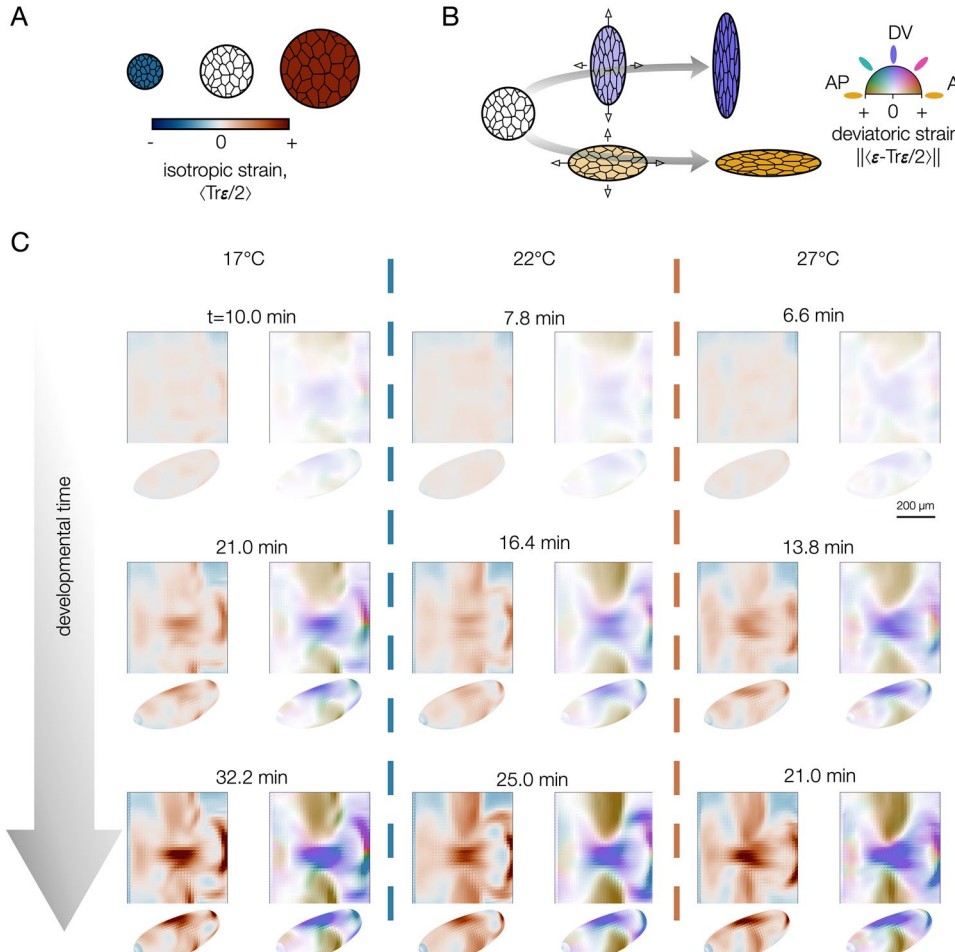

**Extended Data Fig. 4 | Tissue deformation fields match across temperature conditions after a simple rescaling of time.** *(A-B)* Strain fields, which measure tissue deformation, are decomposed into isotropic and deviatoric components. *(A)* The isotropic component of a strain tensor $\varepsilon$ captures the increase (positive) or decrease (negative) in area of a tissue patch. *(B)* The deviatoric component of a strain tensor $\varepsilon$ captures the amount of area-preserving deformation, or convergent extension. The color denotes the axis of elongation of the tissue patch, and the opacity reflects the amount of deformation. *(C)* Ensemble-averaged strain fields during GBE for embryos observed at 17°C (left), 22°C (middle), and 27°C (right) show striking similarity upon rescaling time (embryos are from the same temperature-controlled ensembles as Fig. 5). Note that the elapsed time for embryos in cooler temperatures is shorter than for those in warmer conditions. For each snapshot, the isotropic and deviatoric components are rendered in both 2D pullback projection and 3D surface representation.

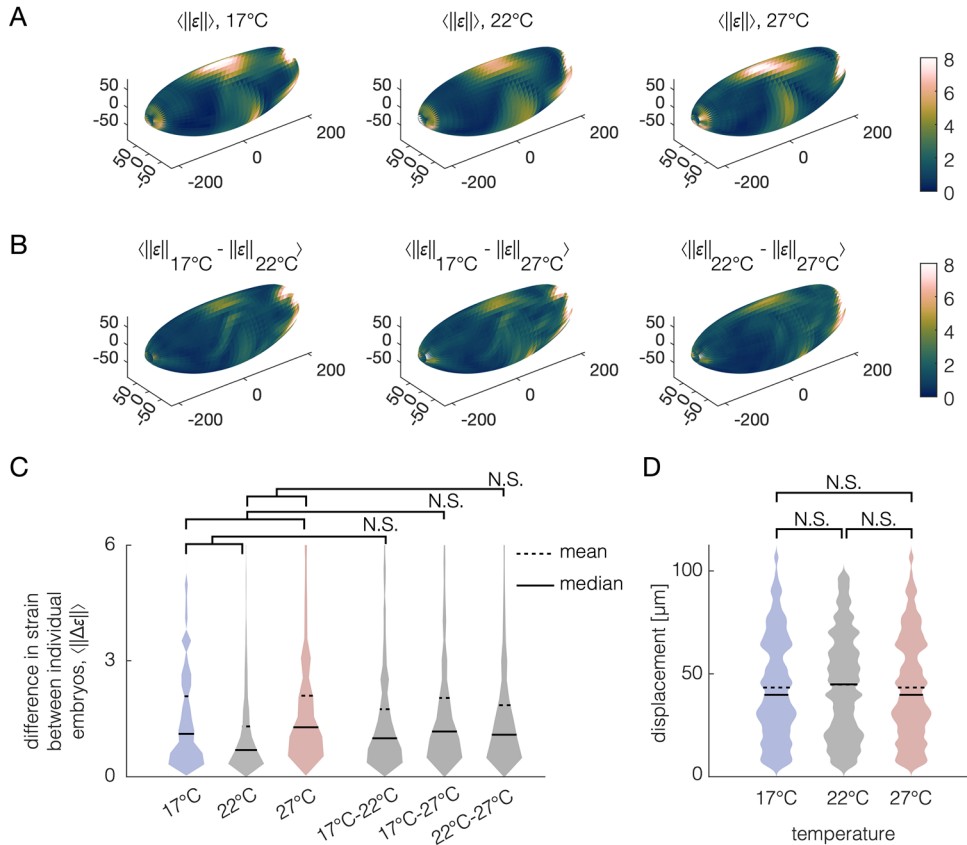

**Extended Data Fig. 5 | Strain fields across temperatures match quantitatively after rescaling time.** *(A)*-*(D)* Computations performed in the same temperature-controlled ensembles as Fig. 5. *(A)* The Frobenius norm of the strain after 32.2 min, 25.0 min, and 21.0 min for 17°C (left), 22°C (middle), and 27°C (right), respectively, show qualitative agreement. The norm is defined as $\| \varepsilon \| \equiv \sqrt{\mathrm{Tr}\varepsilon^2}$. *(B)* In contrast to (A), the mean difference in strain between all pairwise comparisons of spatially-resolved tissue deformations is quiescent, except at the posterior pole where the resolution of our measurements is poor. Here we subtract strain tensors for each individual embryo in one condition from the strain tensors of each embryo in a second condition to compute a spatially-resolved strain difference, and take the Frobenius norm as before. *(C)* The difference in strains across conditions is not significantly different than the difference within the reference conditions. For example, the strain differences between embryos observed at 17°C versus those at 22°C is not significantly different than the collection of strain differences between embryos at 17°C (or at 22°C). Here from left to right, p= 0.235, 0.525, and 0.213 (Kolmogorov-Smirnov tests, single comparisons). *(D)* The displacements of tissue parcels, computed via integration of PIV velocities, did not vary significantly between groups. Kolmogorov-Smirnov tests returned p-values of p= 0.61, 1.00, and 0.26 for comparing 17°C vs 22°C, 17°C vs 27°C, and 22°C vs 27°C, respectively.

A

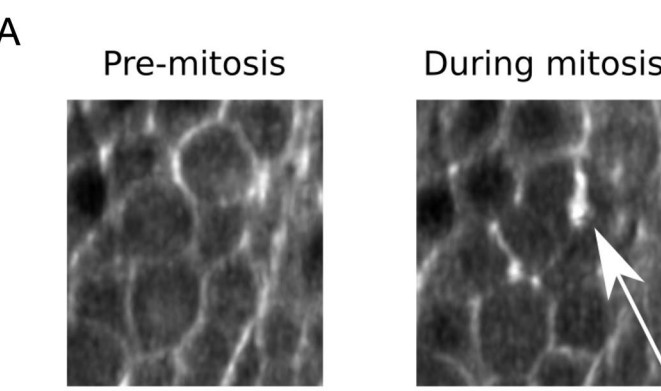

B

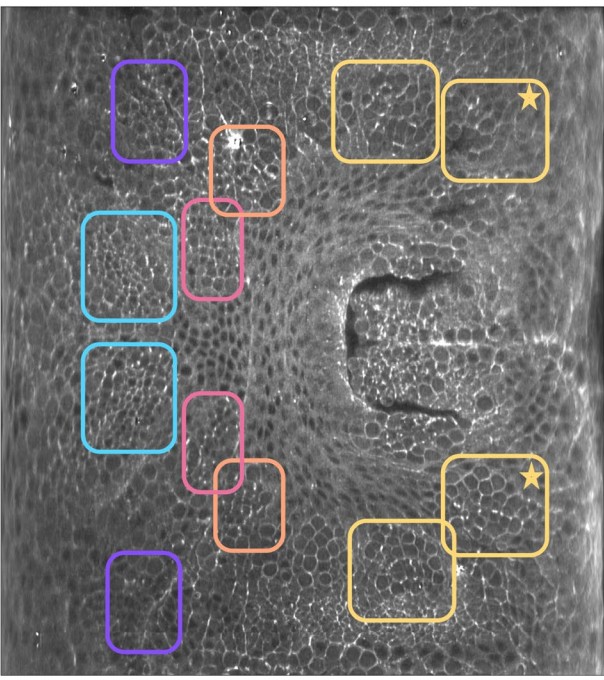

**Extended Data Fig. 6 | Recording of mitotic events, visible as bursts of myosin before cell division, across five mitotic domains.** *(A)-(B)* Wild-type embryo, labeled with a myosin marker (sqh-GFP). *(A)* Example of division event visible using fluorescently-tagged myosin, with myosin flash during cell division marked by a white arrow. *(B)* Mitotic domains {1,2,5,6,11} are respectively color coded in order of appearance. From first to last: cyan, purple, pink, orange, yellow (yellow star denotes a more posterior region of domain 11).

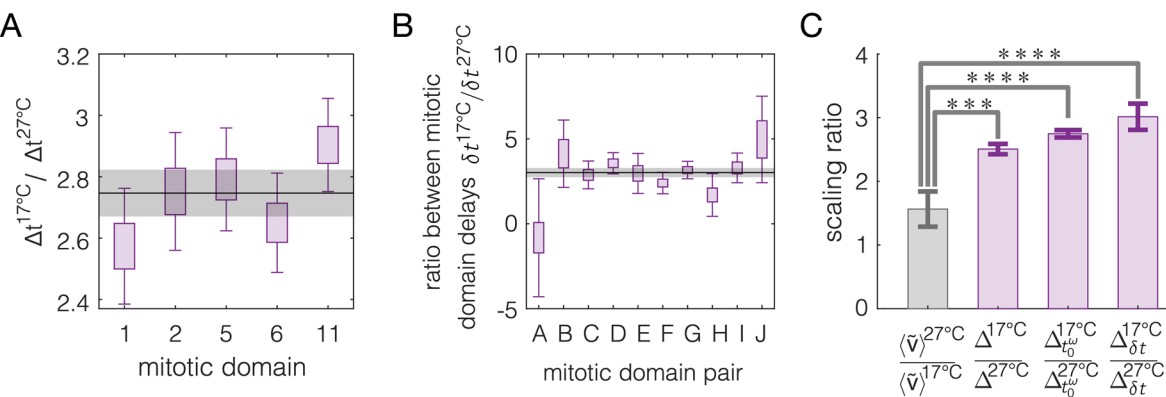

**Extended Data Fig. 7 | Different choices of reference times lead to similar values for the mitotic scaling ratio, which significantly differ from the tissue flow scaling ratio.** *(A)-(C)* Computations performed in the same temperature-controlled ensembles as Fig. 5. *(A)* Ratios of time elapsed since t= 0 (defined as onset of GBE) for 5 mitotic domains. *(B)* Ratios of time elapsed between onsets of division in pairs of mitotic domains. *(C)* Mean scaling ratios across mitotic domains, calculated in three different ways. $\Delta$ without subscript denotes time measured from the onset of GBE to the onset of division, $\Delta_{t_0^{\omega}}$ denotes time measured from the vorticity sign reversal time $t_0^{\omega}$ (see SI Supplementary Note 6) to the onset of division, and $\Delta_{\delta t}$ denotes time between onsets of division in pairs of domains. Here *** denotes $p < 0.001$ and **** denotes $p < 0.0001$ (two-sample t-tests, single comparison).

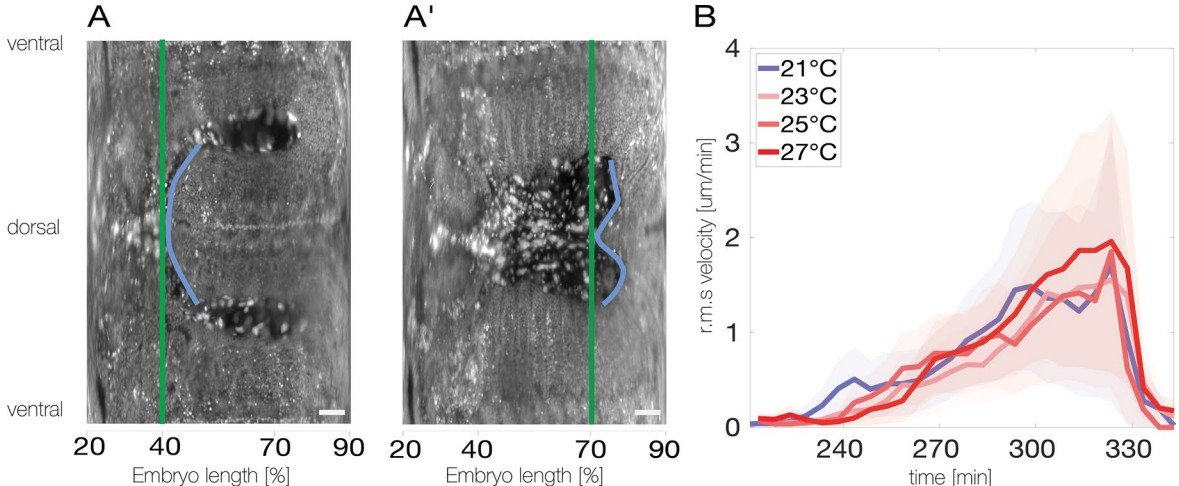

**Extended Data Fig. 8 | GBR tissue flow does not scale with temperature.**
*(A)-(A')* Snapshots of a wild-type embryo undergoing germband retraction (GBR), taken at 21°C. Blue line denotes anterior end of the germband (manually drawn), green line denotes its anterior-most position along the AP-axis. Labeled with a nuclear marker (Histone-RFP). Anterior and posterior poles are truncated. Scale bars 25μm. *(A)* Snapshot during early GBR, with the germband anterior at 40% of the AP-axis. *(A')* Snapshot near the end of GBR, with the germband anterior at 70% of the AP-axis. *(B)* RMS velocity of surface tissue flow during GBR, across temperature conditions. Temperatures 21°C, 23°C, 25°C, 27°C shown (n=1 for all conditions, wild-type embryos). Shading: SD across the surface. Curves aligned such that maximum RMS velocities coincide among all curves. Labeled such that time 0 min corresponds with onset of GBE.

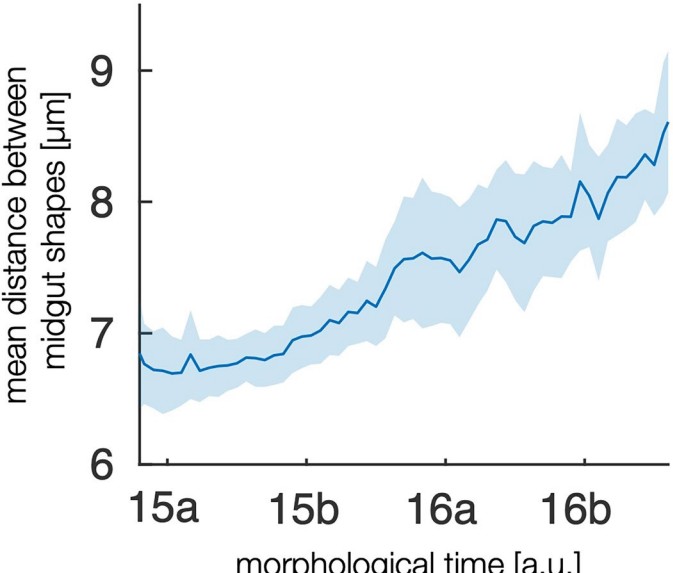

**Extended Data Fig. 9 | Gut morphology increases in variability over morphological time.** Within our ensemble of extracted and spatiotemporally aligned midgut surfaces (n=6 wild-type embryos, same ensemble as Fig. 6c), we measure the mean distance between pairwise gut surfaces. In detail: for each aligned surface pair $s_i(\tau)$ and $s_j(\tau)$ at morphological time $\tau$, we subsample surface $s_i$ and measure the distance to the nearest point in surface $s_j$. Similarly, we subsample $s_j$, and measure the distance to the nearest point in surface $s_i$. We thereby find the mean distance from surface $s_i$ to surface $s_j$ and the mean distance from surface $s_j$ to surface $s_i$. Taking the geometric mean then defines the mean distance between midgut shapes. This distance increases over the timecourse of midgut constrictions, reflecting the increasing uniqueness of each embryo's midgut shape. Shading: SD.

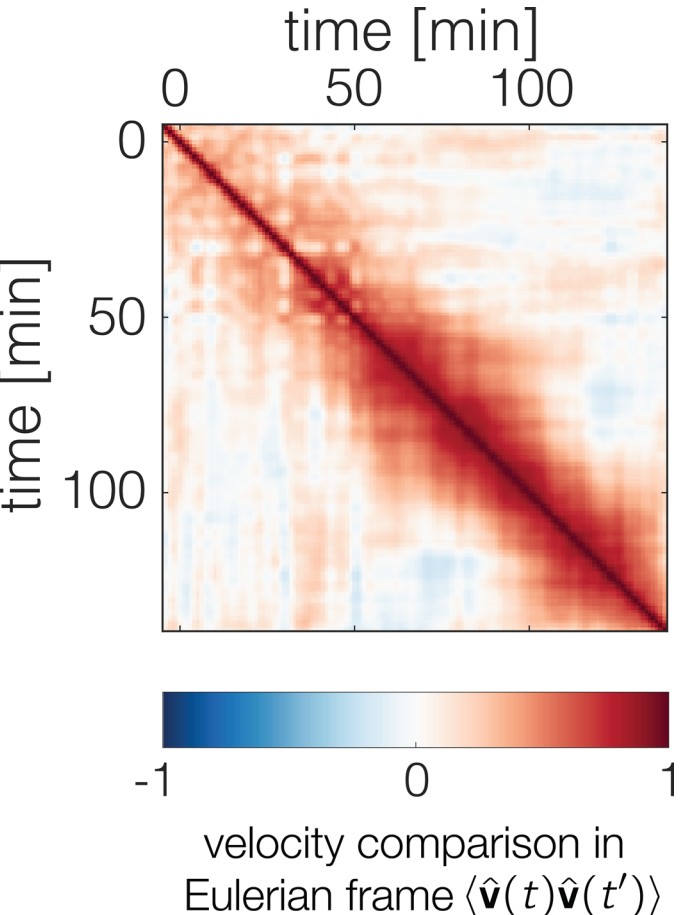

**Extended Data Fig. 10 | Autocorrelation of 3D tissue velocity in the Eulerian (lab) frame of reference for a representative embryo midgut shows little temporal structure.** Autocorrelation matrix of midgut 3D tissue velocity. The midgut analyzed here is the same as in Fig. 6g, but autocorrelation computations are performed in the Eulerian reference frame, rather than in the Lagrangian reference frame used in Fig. 6g. As in Fig. 6g, time axes represent movie acquisition time.

# Reporting Summary

## Statistics

For all statistical analyses, confirm that the following items are present in the figure legend, table legend, main text, or Methods section.

| n/a | Confirmed | |
|---|---|---|
| ☐ | ☒ | The exact sample size ($n$) for each experimental group/condition, given as a discrete number and unit of measurement |
| ☒ | ☐ | A statement on whether measurements were taken from distinct samples or whether the same sample was measured repeatedly |
| ☐ | ☒ | The statistical test(s) used AND whether they are one- or two-sided *Only common tests should be described solely by name; describe more complex techniques in the Methods section.* |
| ☒ | ☐ | A description of all covariates tested |
| ☒ | ☐ | A description of any assumptions or corrections, such as tests of normality and adjustment for multiple comparisons |
| ☐ | ☒ | A full description of the statistical parameters including central tendency (e.g. means) or other basic estimates (e.g. regression coefficient) AND variation (e.g. standard deviation) or associated estimates of uncertainty (e.g. confidence intervals) |
| ☐ | ☒ | For null hypothesis testing, the test statistic (e.g. $F$, $t$, $r$) with confidence intervals, effect sizes, degrees of freedom and $P$ value noted *Give P values as exact values whenever suitable.* |
| ☒ | ☐ | For Bayesian analysis, information on the choice of priors and Markov chain Monte Carlo settings |
| ☒ | ☐ | For hierarchical and complex designs, identification of the appropriate level for tests and full reporting of outcomes |
| ☐ | ☒ | Estimates of effect sizes (e.g. Cohen's $d$, Pearson's $r$), indicating how they were calculated |

*Our web collection on statistics for biologists contains articles on many of the points above.*

## Software and code

Policy information about availability of computer code

| | |
|---|---|
| Data collection | Light sheet imaging data was collected using a custom MuVi SPIM confocal light sheet microscope (see doi: 10.1038/nmeth.2064 for details). For control and automation of the microscope hardware and for development of data capture scripts, the open source software package microManager (https://micro-manager.org/ (version 1.4)) was used. Additionally, the commercial software package Matlab (from release Matlab R2015a to present) was used to generate custom scripts (previously described in: doi: 10.1038/nmeth.2064) to control galvanometric mirrors of the microscope. For fusion and deconvolution of the light sheet data, open source software package ImageJ (https://imagej.net/software/fiji/ (any version released after 2014)) was used with the Multiview Deconvolution plugin (available here: https://github.com/PreibischLab/multiview-reconstruction and described previously doi: 10.1038/nmeth.2929). Additionally, the commercial software package Matlab (from release Matlab R2015a to present) was used to generate cartographic projections of light sheet data as described previously (doi: 10.1038/nmeth.3648). |
| Data analysis | Commercial software used: Matlab R2015a to present release. DynamicAtlas software resources, custom-developed in this publication, were used to perform computational analysis of tissue dynamics, in particular to generate time stamps and perform time alignment across datasets. DynamicAtlas code for the Python-based interface is located on a Zenodo data repository, publicly available at the following URL: "https://doi.org/10.5281/zenodo.14285126" (Version released for publication: v1). DynamicAtlas code for the MATLAB-based interface is located on a Github repository, publicly available at the following URL: "https://github.com/npmitchell/dynamicAtlas" (Version released for publication: v1.1.0). |

For manuscripts utilizing custom algorithms or software that are central to the research but not yet described in published literature, software must be made available to editors and reviewers. We strongly encourage code deposition in a community repository (e.g. GitHub). See the Nature Portfolio guidelines for submitting code & software for further information.

## Data

Policy information about [availability of data](availability of data)

All manuscripts must include a [data availability statement](data availability statement). This statement should provide the following information, where applicable:

- Accession codes, unique identifiers, or web links for publicly available datasets
- A description of any restrictions on data availability
- For clinical datasets or third party data, please ensure that the statement adheres to our [policy](policy)

The experimental data contained within the atlas (including the live datasets used to perform analysis of germband extension) are available on a public Dryad Repository at the following URL: "https://doi.org/10.25349/D9WW43". This includes the spreadsheet of metadata used by the Python interface to query data (MorphodynamicAtlas.csv). We have also included a minimal `demo dataset' (DEMO_DATASET.tar.lz4), used in the MATLAB tutorial contained in the Supplementary Information. This demo dataset is located on a Zenodo data repository, publicly available at the following URL: "https://doi.org/10.5281/zenodo.14792464".

## Human research participants

Policy information about [studies involving human research participants and Sex and Gender in Research.](studies involving human research participants and Sex and Gender in Research.)

| | |
|---|---|
| Reporting on sex and gender | N/A |
| Population characteristics | N/A |
| Recruitment | N/A |
| Ethics oversight | N/A |

Note that full information on the approval of the study protocol must also be provided in the manuscript.

# Field-specific reporting

Please select the one below that is the best fit for your research. If you are not sure, read the appropriate sections before making your selection.

☒ Life sciences          ☐ Behavioural & social sciences          ☐ Ecological, evolutionary & environmental sciences

For a reference copy of the document with all sections, see [nature.com/documents/nr-reporting-summary-flat.pdf](nature.com/documents/nr-reporting-summary-flat.pdf)

# Life sciences study design

All studies must disclose on these points even when the disclosure is negative.

| | |
|---|---|
| Sample size | Our atlas consists of 500 unique live and fixed embryo datasets, including 18 mutant genotypes, detailed on Pages 8-9 of the Supplementary Information in Supplementary Table 1. This sample size was determined by the total number of Drosophila embryos imaged and post processed that passed the quality control standard described below (Data exclusions section). Given the reproducibility of Drosophila tissue dynamics across embryos, this sample size is sufficient to robustly study dynamics of morphogenetic processes across genotypes. |
| Data exclusions | For this study, experimental data inclusion/exclusion decisions were based on pre-established criteria of image and sample quality. All of the data included in this study was acquired on a custom light-sheet microscope. Certain visibly obvious imaging and fusion artifacts (described in detail on Page 11 of the Supplementary Information in Supplementary Note 3) specific to in-toto light sheet microscopy can occur infrequently using the data collection methodologies we employed. To account for these infrequent effects (which visibly and obviously manifest during the multi-view fusion and cartographic surface projection phases of data processing), datasets that did not successfully fuse and/or datasets in which the surface projection was not optimal were excluded from inclusion in the atlas. |
| Replication | For all experimental classes included in the atlas (including different Drosophila mutants analyzed and different fluorescent markers analyzed), a minimum of three independent embryos were included to increase replicability. All attempts at replication were successful. |
| Randomization | Not applicable. This work describes a non-biased atlas of protein expression and tissue flow during Drosophila gastrulation and midgut morphogenesis. Drosophila embryos were segregated into genotypic classes and further segregated by which fluorescently tagged proteins or immunofluorescent targets were used during imaging. |
| Blinding | Blinding is not relevant for this study because we are generating a descriptive atlas of protein expression and tissue flow during Drosophila embryogenesis. This study does not address a particular hypothesis, but rather provides an unbiased dataset of protein expression and tissue flow during Drosophila embryogenesis in wild type and mutant strains. |

# Reporting for specific materials, systems and methods

We require information from authors about some types of materials, experimental systems and methods used in many studies. Here, indicate whether each material, system or method listed is relevant to your study. If you are not sure if a list item applies to your research, read the appropriate section before selecting a response.

## Materials & experimental systems

| n/a | Involved in the study |
|-----|------------------------|
| ☐ | ☒ Antibodies |
| ☒ | ☐ Eukaryotic cell lines |
| ☒ | ☐ Palaeontology and archaeology |
| ☐ | ☒ Animals and other organisms |
| ☒ | ☐ Clinical data |
| ☒ | ☐ Dual use research of concern |

## Methods

| n/a | Involved in the study |
|-----|------------------------|
| ☒ | ☐ ChIP-seq |
| ☒ | ☐ Flow cytometry |
| ☒ | ☐ MRI-based neuroimaging |

## Antibodies

**Antibodies used**

PRIMARY ANTIBODIES: Rabbit anti-Even-Skipped: Gift from Mark Biggin. Guinea Pig anti-Runt: Gift from Wieschaus lab. Mouse anti-Paired: Gift from Nipam Patel (Pax3/7 DP312). Rabbit anti-Sloppy Paired: Gift from Mark Biggin (Rabbit 20257). Rat anti-Hairy: Gift from Wieschaus Lab (Rat 674). Rabbit anti Fushi-Tarazu: Gift from Mark Biggin (Rabbit 11175). Rat anti-Toll6: Gift from Liqun Luo (residues 62-81 RPLTAGAGGDPSLYDAPDDC). Rabbit anti-Tartan: Gift from Wieschaus Lab. Rabbit anti-Bazooka: Gift from Wieschaus Lab. Mouse Anti-Neurotactin: Developmental Studies Hybridoma Bank (Cat # BP106). Rabbit Anti-GFP: Invitrogen (Cat # A11122). Rat Anti-E-cadherin: Developmental Studies Hybridoma Bank (Cat # DCAD2).

SECONDARY ANTIBODIES: Donkey and goat secondary antibodies conjugated to Alexa Fluor 488, 568, 647. Supplied by ThermoFisher Scientific:

Donkey: Alexa Fluor 488 (Catalog # A-21206, RRID AB_2535792), Alexa Fluor 568 (Catalog # A10042, RRID AB_2534017), Alexa Fluor 647 (Catalog # A-31573, RRID AB_2536183)

Goat: Alexa Fluor 488 (Catalog # A-11008, RRID AB_143165), Alexa Fluor 568 (Catalog #A-11011, RRID AB_143157), Alexa Fluor 647 (Catalog # A-21244, RRID AB_2535812)

**Validation**

All antibodies except Rabbit Anti-GFP are generated against Drosophila melanogaster proteins. All antibodies have been validated by the manufacturer, and/or in corresponding publications, as applicable. Listed below (information also contained in Supplementary Table 2 on Page 10 of the Supplementary Information):

GFP (Rabbit, Invitrogen A11122) Validated by manufacturer and referenced in 2052 publications: https://www.thermofisher.com/antibody/product/GFP-Antibody-Polyclonal/A-11122
Even-Skipped (Rabbit, Gift from Mark Biggin, Rabbit #10900) Referenced in Perry et al., Current Biology 2012.
Runt (Guinea Pig, Gift from Wieschaus Lab)
Paired (Pax 3/7 DP312) (Mouse, Gift from Nipam Patel) Referenced in Davis et al., Developmental biology 2005.
Sloppy Paired 1 (Rabbit, Gift from Mark Biggin, Rabbit #20257)
Hairy (Rat, Gift from Wieschaus Lab, Rat #674)
Fushi Tarazu (Rabbit, Gift from Mark Biggin, Rabbit #11175)
Toll-6 (residues 62 to 81 RPLT-AGAGGDPSLY-DAPDDC ) (Rat, Gift from Liqun Luo) Referenced in Ward et al., Neuron 2015.
Tartan (Rabbit, Gift from Wieschaus Lab) Referenced in Lefebvre et al., eLife 2023.
Bazooka (Rabbit, Gift from Mo Weng) Referenced in Gu et al., Molecular Biology of the Cell 2024
Nerotactin (Mouse, DSHB BP106) Referenced in 20 publications: https://dshb.biology.uiowa.edu/BP-106-anti-Neurotactin
E-cadherin (Rat, DSHB DCAD2) Referenced in 48 publications: https://dshb.biology.uiowa.edu/DCAD2

## Animals and other research organisms

Policy information about studies involving animals; ARRIVE guidelines recommended for reporting animal research, and Sex and Gender in Research

**Laboratory animals**

Drosophila melanogaster: Laboratory strain: Oregon R (BDSC (#5)). Data included in the manuscript includes images of Drosophila melanogaster embryos 3-4 hours post fertilization (stages 6-8), 7-9 hours post fertilization (stage 12), and 13-15 hours post fertilization (stages 15-16). Both male and female animals were used; embryo sex could not be determined at these stages. Other strains used are listed below (also listed in Supplementary Table 1 on Pages 8-9 of the Supplementary Information):

UAS-Baz::GFP (Krahn et al., Current Biology 2010)
Klar SqhGFP Tl[rm9] (Gift from Wieschaus Lab)
Klar SqhGFP Spz[4] (Gift from Wieschaus Lab)
sqh1 FRT101/FM7; P{w+ sqh-sqhAE::GFP}attP1 (Gift from Adam Martin; Vasquez et al., Journal of Cell Biology 2014)
Halo [DF2L] snail [IIG05] / CyO, Sqh-GFP (Gift from Adam Martin; Martin et al., Nature 2009)

Halo [DF2L] twist [ey53] / CyO, Sqh-GFP (Gift from Adam Martin; Martin et al., Nature 2009)
P{sGMCA-MoeGFP}on III (Kiehart et al., The Journal of cell biology, 2000)
endo-Ecad::GFP (BDSC (#60584))
w; ubi-DE-cad::GFP (Oda et al, Journal of cell science, 2001)
w; ui-DE-Cad::GFP shg[R69]; Sqh::mCherry[M1] (Gift from Adam Martin; Martin et al., Nature 2009)
yw; sqh-sqh::mCherry[B1] (Gift from Adam Martin; Martin et al., Nature 2009)
w ;; sqh-sqh::mCherry[A11] (Gift from Adam Martin; Martin et al., Nature 2009)
Toll-8::SYFP2 (Gift from Jennifer Zallen; Paré et al, Nature 2014)
Even-Skipped::SYFP2 (Ludwig et al, PLoS genetics 2011)
P{ubi-GFP::rock}/TM3 (Gift from Yohanns Bellaiche; Bardet et al., Developmental cell 2013)
H2A::RFP (Gift from Wieschaus Lab)
H2Av::mCherry (Streichan Lab; Krzic et al., Nature Methods 2012)
sqh-utr::mCherry/ CyO (Gift from Thomas Lecuit; Rauzi et al., Nature 2010)
yw sqh[1] FRT101/FM7; P{w+ sqh-TS::GFP}attP40 (Gift from Adam Martin; Vasquez et al., Journal of Cell Biology 2014)
sqh[Ax3]; P[w+ sqh-gfp]42 (Royou et al., Journal of Cell Biology 2002)
sqh–GFP::ROCK(K116A) (Gift from Jennifer Zallen; de Matos Simões, Developmental cell 2010)
y1 w*; P{UAS-Lifeact::GFP}VIE-260B (BDSC (#35544))
Runt::LlamaTag-GFP (Gift from Hernan Garcia; Bothma et al., Cell 2018)
Tub67c-CAAX::mCherry<sqh3'UTR(attp2)/Tm3,sb (Gift from Wieschaus Lab)
w; 48Y-GAL4; klar (BDSC (#4935), klar from Wieschaus Lab)
w[*]; P{w[+mC]=UAS-mCherry.CAAX.S}2 (BDSC (#59021))
w[*]; UASp-CIBN::pmGFP; UASp-mCherry::CRY2-OCRL (Gift from Stephano de Renzis)
w[*]; UASp-CIBN::pmGFP; UASp-RhoGEF2-CRY2::mCherry (Gift from Stephano de Renzis)
y,P{w[+mC]=GAL4-Antp.P1.A}1,y[1]w[*];wg[Sp-1]/CyO;;klar (BDSC (#26817), klar from Wieschaus Lab)
w;; Mef2-GAL4 klar (Gift from Lucy O'Brian, klar from Wieschaus Lab)
w; UAS-LifeAct::Ruby (BDSC (#35545))
Hand-GFP; 4x HandGAL4; klar (Gift from Zhe Han, klar from Wieschaus Lab)
Bicoid[E1], Nanos[BN] / TM3, sb  (Gift from Wieschaus Lab)
Bicoid[E1], Nanos[BN], Tsl[4] / TM3, sb (Gift from Wieschaus Lab)
Concertina[RC10] cn bw / CyO; T48 p[w+ sqhGFP] (Gift from Wieschaus Lab)
w; ΔJ29, Even-Skipped[r13] / CyO (Gift from Wieschaus Lab)
UAS–Fat2-RNAi (Gift from Sally Horne-Badovinac; Chanet et al., Nature communications 2017)
w; Traffic Jam-GAL4; Gap43::mCherry, sqh::GFP (Gift from Adam Martin; Chanet et al., Nature communications 2017)
UASp–Toll-2–HA (Gift from Jennifer Zallen; Paré et al, Nature 2014)
UASp–Toll8–HA  (Gift from Jennifer Zallen; Paré et al, Nature 2014)
y w; P{UAS-runt.T}15 (Gift from Peter Gergen; Tracey et al., Development 1998)
UAS-Even-skipped / TM6 P[rosy+{l(3)}] (Gift from Andrea Brand; Brand et al., Development 1993)
Dpp[4] Snail[IIG05] / CyO (Gift from Wieschaus Lab)
Dpp[H46] wg[Sp-1] cn[1] bw[1]/CyO, P{dpp-P23}RP1 (BDSC (#2061))
y[1] w[*]; Pmatalpha4-GAL-VP1667; Pmatalpha4-GAL-VP1615  (BDSC (#80361))
w[*]; P{w[+mC]=His2Av-mRFP1}II.2 (BDSC (#23651))

| | |
|---|---|
| Wild animals | No wild animals were used in the study. |
| Reporting on sex | N/A |
| Field-collected samples | No field-collected samples were used in the study. |
| Ethics oversight | Ethics oversight was not required for non-vertebrate studies. All experiments were conducted in the United States, in accordance with the Animal Welfare Act, Health Research Extension Act, and National Institutes of Health regulations. |

Note that full information on the approval of the study protocol must also be provided in the manuscript.

