## [Peer Review File · Nature Methods]

DynamicAtlas: A Morphodynamic Atlas for Drosophila Development

Corresponding Author: Professor Sebastian Streichan

Version 0:

Decision Letter:

2nd Dec 2024

Dear Sebastian,

Your Resource, "DynamicAtlas: A Morphodynamic Atlas for Drosophila Development", has now been seen by 3 reviewers. As you will see from their comments below, although the reviewers find your work of considerable potential interest, they have raised a number of concerns. We are interested in the possibility of publishing your paper in Nature Methods, but would like to consider your response to these concerns before we reach a final decision on publication.

We therefore invite you to revise your manuscript to address these concerns especially the code. In addition, we think the analyses of the atlas needs to be expanded to demonstrate it's broad utility and relevance. It would also benefit the readability of the paper to explain the jargon.

Link Redacted

We hope to receive your revised paper within ten weeks. If you cannot send it within this time, please let us know. In this event, we will still be happy to reconsider your paper at a later date so long as nothing similar has been accepted for publication at Nature Methods or published elsewhere.

OPEN SCIENCE REQUIREMENTS

REPORTING SUMMARY AND EDITORIAL POLICY CHECKLISTS

DATA AVAILABILITY

All novel DNA and RNA sequencing data, protein sequences, genetic polymorphisms, linked genotype and phenotype data, gene expression data, macromolecular structures, and proteomics data must be deposited in a publicly accessible database, and accession codes and associated hyperlinks must be provided in the "Data Availability" section.

CODE AVAILABILITY

Please include a "Code Availability" subsection in the Online Methods which details how your custom code is made available. Only in rare cases (where code is not central to the main conclusions of the paper) is the statement "available upon request" allowed (and reasons should be specified).

MATERIALS AVAILABILITY

More details about our materials availability policy can be found at <https://www.nature.com/nature-portfolio/editorial->

policies/reporting-standards#availability-of-materials

ORCID

Sincerely,
Madhura

Madhura Mukhopadhyay, PhD
Senior Editor
Nature Methods

Reviewers' Comments:

Reviewer #1 (Remarks to the Author):

This paper describes the construction of a resource which integrates both fixed and live imaging data in a single atlas of *Drosophila* development, during the highly studied process of germband extension. Like an earlier gene expression atlas (published by the Berkeley *Drosophila* Transcription Network Project), the expression data here is aligned partly by co-staining with a pair-rule gene (*Runt*). However, the current atlas uses live imaging, with a fluorescently tagged *Runt*, to co-align data based on only tissue deformation. The data and software needed to access this atlas are provided (though see comments below).

To show that their approach for creating a developmental atlas can be generalized to tissues that change shape as they develop, the authors also described and demonstrated a procedure for extracting the surface of the *Drosophila* midgut throughout its development, and mapping individual images to a consensus timeline. Although a more complete atlas of this tissue (complete with gene expression patterns) was not developed in this paper, this is an interesting proof-of-concept that the needed spatial-temporal alignments can be performed.

Although the resource described in this paper could be useful in principle, it needs more work as it currently stands to be of broad utility to the field. One deficit is that the instructions for using the code are not sufficient for many of the potential users, who may be biologists with minimal computational expertise. Additionally, either there is a problem in the code itself, or I was myself unable to follow the directions properly. I closely followed the instructions included in the demo information, cloning the repository, downloading the data from Dryad, and setting the path as described. However, the Matlab code in "demo_dynamicAtlas_functionality.m" did not run properly for me. (Note that I ran this using MATLAB R2024b, on a Mac running Sequoia 15.1.) Instead, I received the following error messages (bracketed by dotted lines):

Insufficient number of outputs from right hand side of equal sign to satisfy assignment.

Error in dynamicAtlas.lookupMap/buildLookupMap (line 61)
genoParentDir = labelDirs.folder ;
AAAAAAAAAAAAAAAAAAAA

Error in dynamicAtlas.lookupMap (line 108)
obj.map = buildLookupMap(obj, options) ;
AAAAAAAAAAAAAAAAAAAA

Error in dynamicAtlas.dynamicAtlas/buildLookup (line 163)
lum = dynamicAtlas.lookupMap(genoDir, Options) ;
AAAAAAAAAAAAAAAAAAAA

Error in dynamicAtlas.dynamicAtlas (line 102)
da.buildLookup(da.genotypes, Options)
AAAAAAAAAAAAAAAAAAAA

Error in demo_dynamicAtlas_functionality (line 33)
da = dynamicAtlas.dynamicAtlas(atlasPath, {'WT'}) ;
AAAAAAAAAAAAAAAAAAAA

It is therefore hard for me to judge the actual utility of this resource.

Reviewer #1 (Remarks on code availability):

I was able to install the code, but could not run it without errors. I am not sure whether there is a problem with the code itself, or whether the README file was insufficiently detailed in its instructions.

Reviewer #2 (Remarks to the Author):

This paper describes a pipeline that generated an atlas that integrates both fixed and live datasets of *Drosophila* embryo development with a single consensus timeline. The key insights described from this dataset include defining specific stages where tissue flow patterns remain unchanging including germband extension. They look at the role of temperature on various aspects of tissue morphogenesis enabled by this morphogenetic atlas pipeline. The authors also investigated a second tissue which is the morphogenesis of the midgut, which is more complex, at a later stage and is a visceral, internal tissue. The explanation of this secondary analysis could be clarified and generalized as the description was somewhat hard to follow compared to the germband extension section.

The strengths of the paper include the ability to generate a high quality and high-resolution consensus atlas with a lot of statistical analysis enabled by the data integration. Overall, the writing is clear; however, some jargon and terminology could be defined more simply to make it more broadly accessible. I enjoyed reading the paper and found the work commendable.

A major weakness is that while several mutants are mentioned in the atlas, almost no analysis was done to test the approach for generating consensus timelines for mutants or the use of the pipeline for generating additional biological insights into those mutant conditions. Instead, the relative 'mild' perturbations of temperature shifts are specified, analyzed and highlighted. As this perturbation does not really impact patterning, the general utility of the pipeline can be questioned.

Points that need clarification:

1. If perturbations impact Runt expression, will the approach easily break down?
2. How many samples are needed to create a robust consensus atlas? Sample size and variability need to be more specifically noted. The rigor and comparison in variability between tissue flow and mitosis needs to be clarified.
3. What is the significance of the reported morphodynamic modules and the mechanism behind the "quasi-stationary" tissue flow patterns? Are there specific hypotheses generated from the analysis?
4. The availability of the large imaging datasets is commendable, but as they are many GBs, it would be useful to include a few compressed video samples as supplementary videos.
5. Will this method work well for late stage of embryogenesis (i.e., during and after dorsal closure?)
6. If the timing of mitotic events varies more than tissue flow velocities, are there significant differences in cell density at each temperature? If not, is there variation in T2 transitions or apoptosis?
7. How can this tool be integrated with other gene expression databases and incorporated into open-source scripts and searching tools?
8. Elements of the pipeline seem to have been utilized previously in other papers, and this could be explained in the paper.

Related to questions of variation and uncertainty, an $n=3$ does not allow for a very precise gauge of variation or uncertainty (even an $n=4$ would be much better) (p. 10)

Misspelling "Cylindrical" p. 9

Why is MATLAB used – would Python enable more accessible and open size and updates?

Why is 0.2 min timestep used for all temperatures?

Enabling a demo with smaller dataset sizes would be useful for testing and evaluation of the approach from outside users.

Does the inclusion of the Runt dataset essential for building the consensus, or could it be done in the absence of gene expression landmarks? How would this framework work in another non model system or stage without nice gene expression pattern data availability?

What are sample sizes in figure 4?

A couple of terms that could be defined more simply: Radon transform (fig. 1), Pullback

Reviewer #3 (Remarks to the Author):

Lefebvre et al. present a morphological atlas that encompasses several stages of *Drosophila* embryonic development. Their atlas is based off of cutting-edge multi-view SPIM imaging that allows an in toto view of the fly embryo. The lab is also expert in segmentation and computational analysis which is performed through customized MatLab coding – these segmented data are then used to generate the atlas. The atlas is compiled from live imaging 495 different fly embryos in 19 different backgrounds (WT and 18 genetically compromised backgrounds). It represents a serious amount of work, and one of the more interesting aspects of their methodology are the approaches that are used to extract cell and tissue flows from so many different embryos that will each have variations in the total size and timing of the processes that occur in them. The study is nice in that they have established elegant solutions to these issues. This is clearly a methods-oriented paper, so, in this respect, is very appropriate for Nature Methods. As the authors acknowledge, it would have been more interesting if the atlas morphologies could have been tied back to changes in gene expression (e.g., RNAseq), but this is admittedly very difficult to do. The MatLab

approaches are also fairly standard in the field, but applied smartly and to a high-level. I therefore think this study could still be interesting to the Nature Methods community. They also show the utility of such an atlas by observing how temperature changes differentially impact different processes in development (gastrulation movements vs cell mitoses), and also nicely show an example of deeper tissue morphological changes, which poses different challenges than surface tissue analyses. I did not find many of the conclusions surprising (outside of the differential impact temp had on mitosis vs morphologies, though why this is the case is unclear) however. The paper is well-written, and data is adequately displayed. I do not have too many critiques; it is largely a well-done study and dataset and to me it just comes down to impact and suitability for the journal. A few further comments...

1) Figures 1 and 2 are very similar flowcharts, it would be nice to try to differentiate these visually as well as provide more methodological details to distinguish them.

2) Standard reviewer complaint that n numbers and statistical tests used should be listed in each figure legend ;-)

3) The authors seem to make a point of commenting that several tissue flows (like germband extension) are highly time dependent and reproducible. Is this interesting? Aren't there many tissues like this? Is there a contrast to be made?

Version 1:

Decision Letter:

Our ref: NMETH-RS57980A

2nd May 2025

Dear Sebastian,

Thank you for submitting your revised manuscript "DynamicAtlas: A Morphodynamic Atlas for Drosophila Development" (NMETH-RS57980A). It has now been seen by the original referees and their comments are below. The reviewers find that the paper has improved in revision, and therefore we'll be happy in principle to publish it in Nature Methods, pending minor revisions to satisfy the referees' final requests and to comply with our editorial and formatting guidelines.

TRANSPARENT PEER REVIEW

ORCID

Sincerely,
Madhura

Madhura Mukhopadhyay, PhD
Senior Editor
Nature Methods

Reviewer #1 (Remarks to the Author):

This paper describes computational methods for integrating fixed and live imaging data into a single atlas, using morphological data. Additionally, this paper describes a very useful resource resulting from these methods: an atlas of *Drosophila* germband extension that incorporates both gene expression data and the dynamics tissue rearrangement, designed as an open source tool to which future datasets can be added. Both the technique and resource described are novel and provide a significant advance to the field. My concern with the previous draft of this manuscript was that I was unable to run the code with the information provided. This concern has been amply addressed: the authors now provide a comprehensive and easy-to-follow tutorial for their Matlab code, as well as a new, simpler Python interface to access the current version of the atlas. These are major improvements, and should make their work broadly accessible to people in the field.

Reviewer #2 (Remarks to the Author):

The work of Lefebvre et al. addressed many points raised in the first review round, which has improved the work. The manuscript also provides a clearer description of interesting insights gained from the quantitative approaches developed.

The work is innovative in its analysis of morphogenesis.

The present work shows that there are differences in scaling with temperature between flow velocity and mitotic times. Second, the work illustrates how patterning directs future modules of morphogenesis at discrete time points, which are characterized by quasi-stationary flow during GBE. This phenomenon depends on DV patterning, suggesting that cross-regulatory principles define constraints on modules. The analysis is then generalized to a second example of 3D tubular morphogenesis.

Question:

For SI Table 1 & 2, can columns be added to clarify if they were used in a specific figure panel or video in the manuscript? The relationship between the resources and data is not clear.

Reviewer #2 (Remarks on code availability):

The SI provides details of tutorial, which aids in understanding the workflow of the code.

Reviewer #3 (Remarks to the Author):

I have no further concerns -- the authors have responded appropriately to reviewer concerns and seriously addressed the critiques. The revised manuscript seems like a good addition to Nature Methods.

Version 2:

Decision Letter:

8th Oct 2025

Dear Sebastian,

I am pleased to inform you that your Resource, "DynamicAtlas: A Morphodynamic Atlas for *Drosophila* Development", has now been accepted for publication in Nature Methods. The received and accepted dates will be Sep 21, 2024 and Oct 08, 2025. This note is intended to let you know what to expect from us over the next month or so, and to let you know where to address any further questions.

Over the next few weeks, your paper will be copyedited to ensure that it conforms to Nature Methods style. Once your paper is typeset, you will receive an email with a link to choose the appropriate publishing options for your paper and our Author Services team will be in touch regarding any additional information that may be required. It is extremely important that you let us know now whether you will be difficult to contact over the next month. If this is the case, we ask that you send us the contact information (email, phone and fax) of someone who will be able to check the proofs and deal with any last-minute problems.

Authors may need to take specific actions to achieve compliance with funder and institutional open access mandates.

If your research is supported by a funder that requires immediate open access (e.g. according to <https://www.springernature.com/gp/open-science/plan-s-compliance>) Plan S principles or the <https://www.springernature.com/gp/open-science/us-federal-agency-compliance> NIH public access policy then you should select the gold OA route, and we will direct you to the compliant route where possible. Because authors warrant under our subscription licensing terms that they haven't committed to licensing any version of their article under a licence

inconsistent with the terms of our agreement – including the applicable embargo period – publication under the subscription model isn't suitable for authors whose funders require no embargo.

If you are active on Twitter/X or Bluesky, please e-mail me your and your coauthors' handles so that we may tag you when the paper is published.

Best regards,
Madhura

Madhura Mukhopadhyay, PhD
Senior Editor
Nature Methods

** Visit the Springer Nature Editorial and Publishing website at http://editorial-jobs.springernature.com?utm_source=ejP_NMeth_email&utm_medium=ejP_NMeth_email&utm_campaign=ejp_Nmeth > www.springernature.com/editorial-and-publishing-jobs for more information about our career opportunities. If you have any questions please click [here](mailto:editorial.publishing.jobs@springernature.com) . **

DynamicAtlas: A Morphodynamic Atlas for Drosophila Development

Point-by-Point Response

Editor:

We therefore invite you to revise your manuscript to address these concerns especially the code. In addition, we think the analyses of the atlas needs to be expanded to demonstrate it's broad utility and relevance. It would also benefit the readability of the paper to explain the jargon.

We thank the editor and the reviewers for the valuable feedback. We have revised our manuscript accordingly.

Following your suggestions, we have made major changes to how we present the atlas code to new users. We have added a Python-based method for interacting with the atlas, which allows a user to query and analyze data using the popular Jupyter Notebook interface. This includes a newly created tutorial notebook that walks through the key Python functions for accessing atlas data (located at <https://zenodo.org/records/14285126>). We anticipate that the vast majority of users will interact with the atlas using this method, and that it will be accessible to users with all levels of computational experience.

For users who wish to perform the heavier-duty Matlab-based functions of the atlas, such as timeline creation and timestamping on a new ensemble, we have improved the presentation in several ways. We have created a small demonstration dataset (located at <https://zenodo.org/records/14792464>) with a corresponding demo script (included as the file "demo_dynamicAtlas_functionality.m" on the DynamicAtlas Github: <https://github.com/npmitchell/dynamicAtlas>) that a user can interact with to test core functions. Paired with this, we have created a new detailed step-by-step tutorial (included as the file "DynamicAtlas_Demo_Tutorial.pdf" on the DynamicAtlas Github: <https://github.com/npmitchell/dynamicAtlas>), including screenshots, to walk through this demo script, and guide the user through the entire pipeline of creating a consensus timeline starting from a set of minimal inputs. Both the Python and the Matlab tutorials linked above are also now attached to the end of the Supplementary Information (Page 30 to 46 and Page 47 to 85).

We anticipate these changes to our code will enable broad utility of our resource to those with all levels of computational experience, and have successfully had our Python and Matlab platforms tested on Mac OS Sequoia 15.1, Windows 10, and Linux Ubuntu 18.04.6, in the hands of users with limited computational experience and no prior experience of the software.

We have also expanded our analysis by demonstrating atlas utility for higher resolution quantitative exploration of mutant ensembles. We have included a new figure, Figure 4, to depict our alignment methods (Figure 4a) and analysis of flow auto-correlations for four mutant ensembles (Figure 4b,c,d,e). The results illustrate that the morphodynamic modules we have found in wild type are genetically regulated, and that DV patterning genes are a prerequisite for establishing their quasi-stationary flows. The figure, and the corresponding additions to the main text on Page 5, detail strategies for overcoming the challenges of analyzing mutant ensembles as compared to wild type.

We have also expanded our analysis in Figure 3 by adding panels 3F-3I, directly showing the contrast between gene expression patterns and flow patterns, which explicitly illustrates the biological context of the flow modules. Additionally, we have more clearly delineated the difference between Figure 1 (graphical overview) and Figure 2 (methods of time alignment).

We have also explained jargon, edited the phrasing of terms, and more clearly referenced existing resources, to enhance the readability of the paper.

We hope these changes further the understanding of the utility and relevance of atlas to a broad audience. A detailed point-by-point response to reviewer points, with corresponding changes noted, follows below. Our responses are in magenta. Copies from the main text addressing reviewer points are reproduced in italics from the revised manuscript, in different styles: black denotes unchanged text, blue denotes added text, and strikethroughs denote removed text.

To Reviewers:

We have revised our main text in reference to both specific and general points raised by the editor and reviewers. Some text additions (in blue) address particular reviewer points, and these are quoted from the revised manuscript in our corresponding responses below. Other text changes rewrite or augment existing text to address the broader feedback to improve clarity and readability; for context, any rewritten/removed prior text is still shown, but struck through. Finally, all figure legends have been revised to improve clarity, and the Methods section has been expanded — both to include methods of new analyses performed (in revised Figures 3, 4, S12) and to explain some existing analyses in greater detail.

Reviewer #1 (Remarks to the Author):

This paper describes the construction of a resource which integrates both fixed and live imaging data in a single atlas of *Drosophila* development, during the highly studied process of germband extension. Like an earlier gene expression atlas (published by the Berkeley *Drosophila* Transcription Network Project), the expression data here is aligned partly by co-staining with a pair-rule gene (*Runt*). However, the current atlas uses live imaging, with a fluorescently tagged *Runt*, to co-align data based on only tissue deformation. The data and software needed to access this atlas are provided (though see comments below).

To show that their approach for creating a developmental atlas can be generalized to tissues that change shape as they develop, the authors also described and demonstrated a procedure for extracting the surface of the *Drosophila* midgut throughout its development, and mapping individual images to a consensus timeline. Although a more complete atlas of this tissue (complete with gene expression patterns) was not developed in this paper, this is an interesting proof-of-concept that the needed spatial-temporal alignments can be performed.

Although the resource described in this paper could be useful in principle, it needs more work as it currently stands to be of broad utility to the field. One deficit is that the instructions for using the code are not sufficient for many of the potential users, who may be biologists with minimal computational expertise. Additionally, either there is a problem in the code itself, or I was myself unable to follow the directions properly. I closely followed the instructions included in the demo information, cloning the repository, downloading the data from Dryad, and setting the path as described. However, the Matlab code in "demo_dynamicAtlas_functionality.m" did not run properly for me. (Note that I ran this using MATLAB R2024b, on a Mac running Sequoia 15.1.) Instead, I received the following error messages (bracketed by dotted lines):

Insufficient number of outputs from right hand side of equal sign to satisfy assignment.

Error in dynamicAtlas.lookupMap/buildLookupMap (line 61)
genoParentDir = labelDirs.folder ;
^^^^^^^^^^^^^^^^^^^^

Error in dynamicAtlas.lookupMap (line 108)
obj.map = buildLookupMap(obj, options) ;
^^

Error in dynamicAtlas.dynamicAtlas/buildLookup (line 163)
lum = dynamicAtlas.lookupMap(genoDir, Options) ;
^^

Error in dynamicAtlas.dynamicAtlas (line 102)

da.buildLookup(da.genotypes, Options)

^^

Error in demo_dynamicAtlas_functionality (line 33)

da = dynamicAtlas.dynamicAtlas(atlasPath, {'WT'});

^^

It is therefore hard for me to judge the actual utility of this resource.

Reviewer #1 (Remarks on code availability):

I was able to install the code, but could not run it without errors. I am not sure whether there is a problem with the code itself, or whether the README file was insufficiently detailed in its instructions.

We apologize for this inconvenient experience working with our code, and for our lack of clarity in our instructions. We greatly appreciate the feedback to make changes so that our atlas is accessible for users of all computational backgrounds, and this perspective helped us understand the clear deficiencies in our presentation, particularly the README on our DynamicAtlas GitHub (now updated: <https://github.com/npmitchell/dynamicAtlas/blob/master/README.md>).

Accordingly, we have now made major changes to how we present the atlas software to a new user. In particular, we have now delineated two separate interfaces that are tailored for the two types of users we anticipate:

- (1) users who primarily wish to access the atlas data as it currently exists (Python interface: <https://zenodo.org/records/14285126>)
- (2) users who wish to add their own data to the atlas and analyze new ensembles of data (Matlab interface: <https://github.com/npmitchell/dynamicAtlas>)

For (1), we have compiled a master spreadsheet of the properties of the data currently in our atlas in a CSV file format (uploaded to our updated Data Dryad repository: <https://datadryad.org/stash/dataset/doi:10.25349/D9WW43>), and written a Python-based interface (located at: <https://zenodo.org/records/14285126>) to explore properties of these data using this CSV as a guide, via the interactive Jupyter notebook environment. We have included in the Supplementary Information (Page 30 to 46) a tutorial Jupyter notebook that enables the user to view or analyze data of interest (Tutorial also located at: <https://zenodo.org/records/14285126>, and CSV located at: <https://datadryad.org/stash/dataset/doi:10.25349/D9WW43>). We anticipate that most users will interact with the atlas using this method, and that it will be accessible at all levels of computational experience. We have successfully had our Python software tested on Mac OS Sequoia 15.1, Windows 10, and Linux

Ubuntu 18.04.6, in the hands of users with limited Python experience and no prior experience of the software.

For (2), we have improved our Matlab-based atlas presentation. We have made our demo script more user-friendly (included in “demo_dynamicAtlas_functionality.m” on the DynamicAtlas Github: <https://github.com/npmitchell/dynamicAtlas>), and we have assembled a smaller demonstration dataset with minimal inputs (located at <https://zenodo.org/records/14792464>) to use with the script. Paired with these, we have now created a detailed tutorial walkthrough, with screenshots — included on the DynamicAtlas Github (<https://github.com/npmitchell/dynamicAtlas>, tutorial file name “DynamicAtlas_Demo_Tutorial.pdf”) and in the Supplementary Information (Page 47 to 85). This performs the key atlas functions, namely for morphological time alignment. We have successfully had our Matlab software tested on Mac OS Sequoia 15.1, Windows 10, and Linux Ubuntu 18.04.6, in the hands of users with no prior experience of the software and limited general computational experience.

We also describe in our new tutorials (Python: <https://zenodo.org/records/14285126> Matlab: <https://github.com/npmitchell/dynamicAtlas> in file “DynamicAtlas_Demo_Tutorial.pdf”) that the atlas code requires certain data formatting conventions — i.e. for folder structure, and file names/formats. Our previous presentation was unclear on this, and we have now explained this in more detail.

Lastly, we were able to reproduce the described Matlab error. The source appears to be as follows: the atlas requires a data folder structure with the parent folder as the genotype, and the child folders as the dataset IDs. If the folders have a different structure, the described error is reproducibly returned. It seems this can occur while compressed data is unzipped, depending on the unzipping program. We are grateful this error was pointed out, and we have clarified how to rectify this in the Matlab tutorial (Supplementary Information, Page 51; Tutorial Page 5).

We thank the reviewer for the patience and productive feedback on our code, and hope our changes will make the atlas accessible for everyone.

Reviewer #2 (Remarks to the Author):

This paper describes a pipeline that generated an atlas that integrates both fixed and live datasets of *Drosophila* embryo development with a single consensus timeline. The key insights described from this dataset include defining specific stages where tissue flow patterns remain unchanging including germband extension. They look at the role of temperature on various aspects of tissue morphogenesis enabled by this morphogenetic atlas pipeline. The authors also investigated a second tissue which is the morphogenesis of the midgut, which is more complex, at a later stage and is a visceral, internal tissue. The explanation of this secondary analysis could be clarified and generalized as the description was somewhat hard to follow compared to the germband extension section.

The strengths of the paper include the ability to generate a high quality and high-resolution consensus atlas with a lot of statistical analysis enabled by the data integration. Overall, the writing is clear; however, some jargon and terminology could be defined more simply to make it more broadly accessible. I enjoyed reading the paper and found the work commendable.

A major weakness is that while several mutants are mentioned in the atlas, almost no analysis was done to test the approach for generating consensus timelines for mutants or the use of the pipeline for generating additional biological insights into those mutant conditions. Instead, the relative 'mild' perturbations of temperature shifts are specified, analyzed and highlighted. As this perturbation does not really impact patterning, the general utility of the pipeline can be questioned.

We thank the reviewer for raising these points. We have reworked our presentation of the midgut figure (now Figure 6) by adding explanatory panels 6E and 6F to more clearly convey the quantitative midgut analysis. Our figure legend for Figure 6 also now includes more detailed explanations of the content shown.

Overall, to enhance the readability of the paper, we have explained jargon and edited the phrasing of terms, see response to point 17.

We have now included a new figure for mutant analysis, Figure 4, which depicts four mutant ensembles treated using our methods (Figure 4a) and analysis of flow auto-correlations (Figure 4b,c,d,e). This figure demonstrates the utility of our atlas for mutant ensembles, and illustrates that the wild-type modules found in Figure 3C are genetically regulated by DV patterning genes.

See added section on Page 5, reproduced below:

Mutant analysis suggests dorsal-ventral patterning is a prerequisite for quasi-stationary flows

Mutant analysis often characterizes a phenotype in terms of the final tissue shape. The atlas introduces higher resolution quantitative methods to this approach. Within a genotype, we assemble an atlas as before. For time alignment across genotypes, we rely on landmarks (Fig. 4A and Methods: Mutant time alignment).

We illustrate this approach by analyzing how genetics affects stationary flow modules. We choose the zygotic mutant eve^{R13} and the maternal double mutant $bcd^{E1}nos^{BN}$ for anterior posterior (AP) patterning (Fig. 4B,C). Both AP mutants are time aligned to WT based on ventral furrow timing. As expected, eve^{R13} retains DC and VF modules. In eve^{R13} , the germband extends less [35], but we find the GBE flow profile remains stationary (Fig. 4B). Consistent with earlier descriptions, $bcd^{E1}nos^{BN}$ exhibits DC and VF modules.

We find a GBE module that is weakened compared to WT (Fig. 4C), likely due to slower overall flow [35].

We choose the zygotic mutant twi^{ey53} and the maternal mutant spz^A for dorsal ventral (DV) patterning (Fig. 4D,E). Both DV mutants are time aligned to WT based on cephalic furrow timing (see Methods: Mutant time alignment). As expected, twi^{ey53} does not exhibit separate DC and VF modules, but retains a stationary GBE flow module (Fig. 4D). In spz^A , we find a complete loss of DC, VF, and GBE modules (Fig. 4E), likely due to near complete cessation of flow. We conclude that the AP patterning mutants retain early WT modules, and exhibit a stationary GBE flow pattern. In contrast, intact DV patterning is required for all stationary flow patterns.

Points that need clarification:

1. If perturbations impact Runt expression, will the approach easily break down?

Thank you for pointing out a better explanation of this critical point. Our time alignment methods do not require Runt expression. We have modified our presentation of Figure 2 to make this more clear. In panels 2D-D'' which have now been added to Figure 2, we illustrate that we can align embryos using integrated tissue flow even if Runt is not present. We describe this method in Section 3:

'Aligning live datasets based on tissue deformation' on Page 3, which we have now revised, see text reproduced below:

Aligning live datasets based on tissue deformation

Embryonic tissue deforms during morphogenesis, and its instantaneous rate of deformation can be captured by flow fields [24] (Fig. 2D). The degree of total deformation can serve as a benchmark for defining morphological time, and can be reconstructed from the instantaneous flow fields (see SI: Instantaneous Flow).

We developed a method to compare integrated flow patterns, cross correlating displacements of the tissue to one another using the same fast marching technique as in aligning live imaging of Runt stripes. In both cases, the morphology of the tissue marks its placement in the morphological timeline. We fix the reference timepoint of the integration as the time when GBE tissue flow starts to rise significantly (see SI: Time-alignment for early embryo datasets). As shown in Fig. 2D-D" and SI Fig. S2, this approach leads to aligned tissue morphology.

Note that we use total tissue deformation, and not the instantaneous flow field, as a developmental landmark. In many contexts, instantaneous flow fields can be relatively constant in time (e.g. cells migrating with fixed speed), making them unsuitable as timestamps.

This method time aligns distinct embryos from tissue deformation alone, and does not require analysis of gene expression patterns. It is possible that this method could be used in other living systems.

We also illustrate in Figure 4 that we can align ensembles of mutants that strongly perturb Runt, such as $bcd^{E1}nos^{BN}$ (Figure 4C), by using tissue flow. Additionally, we show in Figure 6 that we are able to align midguts based on surface shape, without requiring gene expression patterns. See response to point 5, quoting from main text Page 6, for the revised text section describing midgut alignment.

**2. How many samples are needed to create a robust consensus atlas?
Sample size and variability need to be more specifically noted. The rigor and comparison in variability between tissue flow and mitosis needs to be clarified.**

We thank the reviewer for pointing out these clarifications. We have clarified sample numbers, particularly in figure legends. We have noted the sample sizes used in Figure 5, where the comparison between tissue flow and mitosis is made (n=3 for the 17°C and 27°C ensembles). We have also specifically noted the statistical significance of the comparison in variability between tissue flow and mitosis in the Figure 5 legend.

To help readers assess time-stamping robustness, we modified the main text. See addition to Section 2: 'Aligning fixed datasets' on Page 3, reproduced below:

As shown in Fig. 2C, we designed our atlas such that all fixed embryos are multiply immunostained against both Runt and a second gene product target of interest. For each embryo, we extract the static geometry of Runt stripes to correlate its position against the consensus Runt stripe shapes from live datasets (Fig. 2C'), thereby determining the timestamp at which the fit residual is minimum (Fig. 2C"). This process uses stripe geometry as a stopwatch against which we timestamp all fixed samples in the atlas. Any co-stained targets of interest are therefore aligned in time based on the position of the Runt stripe. For our consensus timeline based on live Runt nanobody data (n=5), we find that this method timestamps the average fixed sample to the timeline with uncertainty of +/- 2 minutes (Methods: Uncertainty of Morphological Timestamping). For other ensembles, the timestamping uncertainty can be similarly determined empirically. In general, an ensemble requires a minimum of 3 live samples to perform the timestamping, and the precision of this process can be increased to a desired level by increasing the number of live samples recorded.

3. What is the significance of the reported morphodynamic modules and the mechanism behind the “quasi-stationary” tissue flow patterns? Are there specific hypotheses generated from the analysis?

We think this is an important aspect, and have now highlighted that this effect suggests a novel role of DV patterning. See additions to the Discussion on Page 7, reproduced below:

Quasi-stationary flows in the early Drosophila embryo provide a simple physical way to establish complex shape

changes. Despite the complexity of the embryo, with order 10,000 cells and 10,000 genes [51], we find that the embryo executes the same flow pattern for an extended time. This stationary flow pattern is established by active processes in cells, but it does not change while cells move through it. Instead, the flow rapidly changes once the tissue has achieved a desired geometry.

The adherence of ~~tissue flow~~ the GBE flow pattern to a fixed, geometric frame is reminiscent of the well-documented spatial precision of gene expression patterns in the early blastoderm [52, 53]. These earlier studies showed gene expression levels to be spatially patterned with single-cell precision along the body axes. Tissue kinematics are likewise strongly correlated in a fixed, geometric (Eulerian) reference frame the fixed reference frame of the egg, despite massive motion of the material. This presents a challenge: how does genetic signaling in cells instruct the directions of flow, given that the flow pattern remains aligned with the principal axes of the egg, while cell positions — and thereby expression patterns of nuclear gene concentration — continuously change? The global distribution of myosin is a predictor of tissue flow during GBE [24]. Stationary GBE flow patterns suggest that myosin patterns are also stationary in the body axis frame. This result is in line with the recent finding that the global myosin pattern appears controlled by static cues that align with the DV axis during GBE tissue motion [30]. At the cellular level, recent work has showed cell-cell interfaces preferentially replenish their myosin level according to their degree of DV axis alignment [54]. We find that DV, but not AP patterning is required for stationary flow patterns. Taken together, this adds to the growing body of evidence that the DV patterning system coordinates robust morphogenetic movements of GBE, e.g. by patterning mechanical feedback [55] [56].

4. The availability of the large imaging datasets is commendable, but as they are many GBs, it would be useful to include a few compressed video samples as supplementary videos

We have assembled and included selected downsampled movies in the supplementary material for easy visualization of the type of live data that is contained in this atlas (Supplementary Videos 1, 2, 3). We have also assembled a

smaller demonstration dataset (located at <https://zenodo.org/records/14792464>) that users can use to test our time alignment and time stamping methods, as part of the Matlab code tutorial (<https://github.com/npmitchell/dynamicAtlas> in file "DynamicAtlas_Demo_Tutorial.pdf").

5. Will this method work well for late stage of embryogenesis (i.e., during and after dorsal closure?)

We have both added a new supplementary figure SI Fig. S12, as well as revised the presentation of Figure 6, to demonstrate the analysis of flow close to the surface of the embryo during a later stage.

In SI Fig. S12, we have analyzed the stationary germband retraction (GBR) module under four different temperature conditions (21°C, 23°C, 25°C, 27°C). We have used the alignment and analysis methods of the atlas to study the flow on the surface of the embryo at this later stage. This analysis demonstrates that the scaling behavior we observe during the GBE module does not occur in the GBR module. We have added text to the temperature section on Page 6 accordingly, reproduced below:

In contrast, the stationary GBR module does not scale with temperature (SI Fig. S12). Notably, unlike GBE, GBR is not characterized by cell intercalations in the germband, but by mechanical coupling to the amnioserosa [41][42].

We further have Figure 6, which shows our method works well at late stages of embryogenesis in the context of folding in the embryonic midgut (stages 15-16b). This process occurs multiple hours after the completion of dorsal closure (stage 13). Our analysis here illustrates that the atlas methods can be used even in the context of a complex, dynamic surface. See discussion of spatiotemporal midgut alignment on Page 6, reproduced below:

*With sequences of each embryo's midgut surface in hand, we find the closest match between each pair of surfaces using iterative closest point registration (see Methods: Spatiotemporal alignment of midgut morphogenesis). This algorithm finds the combination of rotation and translation that best maps two 3D surfaces onto each other. This morphological approach allows alignment across embryos ~~with different~~ *independent of* fluorescently-tagged protein (Fig. 6B-B').*

With spatial registration performed, we can use a quantitative comparison of organ shape to temporally

align the process of midgut morphogenesis across embryos. Performing pairwise alignment across embryos leads to a consensus timeline of morphology shown in Fig. 6C. For our ensemble of embryos, the rate of development through morphological stages varied by ~10%. Fig. 6D shows cross-sections of these embryos in the lateral plane during four stages of development. The match between midgut shapes becomes less stereotyped as morphogenesis proceeds, as evident in quantification of the shape variation across our ensemble (Fig. 6D and SI Fig. S3).

Together, this establishes that the method applies to a broad range of stages during *Drosophila* embryogenesis, from surface flows, to the shaping of complex inner organs.

6. If the timing of mitotic events varies more than tissue flow velocities, are there significant differences in cell density at each temperature? If not, is there variation in T2 transitions or apoptosis?

That's a great question. We have modified the temperature section in the main text on Page 5 and Page 6, reproduced below:

To tune the rate of development, we measured the pattern of flow during GBE at a series of temperatures (17°C (n=3), 22°C (n=5), 27°C (n=3). ~~with embryos viable at all temperatures~~ We note that embryos were viable at all temperatures, and the integrity of the epithelium was not affected by the perturbation of temperature (we did not observe noticeable differences in cell densities or apoptosis). As shown in the top of Fig. 5A, embryos cultured at 17°C demonstrated reduced tissue flow rate, while embryos at 27°C progressed through GBE more rapidly. Embryos cultured at 22°C adopted an intermediate flow rate. We find that quasi-stationary kinematics appear in each temperature condition. Moreover, temperature does not affect the spatial pattern of tissue motion encoded by the kinematics: the flow fields differ only in their magnitude.

...

To measure the rate of the 'mitotic clock' at each perturbed temperature (17°C and 27°C), we calculated the

time elapsed between the onsets of division in different mitotic domains (Fig. 5D-E, SI Fig. S8). Remarkably, we find that the ratio of the division time differences at different at the two temperatures diverged significantly from the ratio of maximum flow speeds which characterizes the parameter-free scaling of the tissue motion (Fig. 5D). In particular, Fig. 5D-E shows that the relative timing of mitotic events across conditions varied by a factor of 2.5 +/- 0.1. In contrast, tissue flow velocities varied only by a factor of 1.5 +/- 0.3. The discrepancy between division times and flow rates is robust across different methods of measuring the time difference, such as choosing a different reference time (see SI Fig. S9). This result opens further questions: why do cell cycle and tissue deformation timings scale differently, and how are these differences accommodated to produce viable embryos over a large range of temperatures?

During the time period we are measuring in the temperature-perturbed samples (fast and slow phases of germband extension), we have not noted T2 transitions or apoptosis events, and are unaware of any such events that have been previously described. Our measurements occur right after the conclusion of the 14th syncytial division cycle, which takes place across the temperatures we measure (17°C, 22°C, 27°C). Therefore, our initial condition contains the same cell density, and we would only observe variations starting during the later mitotic domains. However, few cells divide in each domain compared to the number in the overall embryo. And the precise time when each domain undergoes division is not as synchronized as the early divisions, but rather is distributed over times comparable to germband extension as a whole. Therefore we conclude cell density is at most moderately affected by the temperature perturbations.

7. How can this tool be integrated with other gene expression databases and incorporated into open-source scripts and searching tools?

In the current version of the atlas the code is available as an open source resource (on Github (Matlab): (<https://github.com/npmitchell/dynamicAtlas> and Zenodo (Python): <https://zenodo.org/records/14285126>) as is all of the raw data (available on Dryad: <https://datadryad.org/dataset/doi:10.25349/D9WW43>). However it would be very useful to have this data linkable from other bioinformatic databases like FlyBase. We have now reached out to FlyBase for this, and they have expressed interest in integrating our resource with their platform. We are currently working with FlyBase to have our Dryad data repository listed on the FlyBase 'Online Resources' wiki page that links to external atlases (https://wiki.flybase.org/wiki/FlyBase:Drosophila_Online_Resources#Atlases.2C_Im

ages_and_Videos), and to have a README uploaded to the wiki from FlyBase describing how to navigate our resource data & software.

8. Elements of the pipeline seem to have been utilized previously in other papers, and this could be explained in the paper.

In the main text we have references to elements of the pipeline that have been utilized in other papers. We tried to avoid self-referencing before. Unlike the popular light sheet microscopy field, we are one of the few labs developing these specific data analysis techniques. For example, tissue cartography on Page 2, reproduced below:

Our atlas software automatically aligns a cartographic projection of the 3D embryo surface with the embryo body axes for this purpose (using cartography techniques previously developed in [24] and [25], see Methods: Tissue Cartography).

and midgut surface extraction on Page 6, reproduced below:

*To test this, we here extend our spatiotemporal registration and examination of velocity correlations during morphogenesis to the development of complex shapes in the midgut. We use level sets approaches of the TubULAR package (a method developed previously in [47, 49]) to extract a surface that penetrates ~2.5 μm within the apical surface of the endoderm, along the surface that intersects endodermal nuclei. In contrast to the case of ~~germ-band~~ *germband* extension, these surfaces are highly dynamic, demanding additional steps for spatial registration.*

We have also included corresponding references for midgut deformation computation in the added section 'Methods: Correlation of Midgut Deformations' on Page 12, reproduced below:

Correlation of Midgut Deformations

In midgut datasets, out-of-plane deformations were computed and used to calculate tissue-frame (Lagrangian) correlations. Out-of-plane deformations were measured and mapped to 2D surface projections using the methods in [46]. Correlation between projections was then computed as in the Correlation of Images section above.

We hope this will improve the clarity for a reader on which elements of dynamicAtlas are new, and which are derived from existing resources.

9. Related to questions of variation and uncertainty, an n=3 does not allow for a very precise gauge of variation or uncertainty (even an n=4 would be much better) (p. 10)

Thank you for raising these points. To clarify the point about variation, see our response to point number 2 about timestamping robustness. To clarify the point about uncertainty, we have added the following text to the new Methods section: Uncertainty of Morphological Timestamping on Page 13, reproduced below:

The uncertainty of timestamping onto a particular timeline depends on the reproducibility of the morphological feature used for alignment, and the number of samples (minimum $n = 3$ required). For the timelines in our analysis, we note that our time alignment is based on live movies of a highly reproducible process. Each dataset in our $n = 3$ ensembles (e.g. our 17°C and 27°C ensembles in Fig. 5) contains on the order of 50 to 100 datapoints, one per frame. Our alignment method uses all timepoints in all embryos, and therefore uses at least an order of magnitude more datapoints than the number of samples. Because of this quantity of information, we can use the live alignment to create a robust morphological timeline, which can then incorporate many more samples, including fixed datasets (Fig. 1E).

Additionally, we clarify the sample sizes for our temperature ensembles, which we did not clearly specify on Page 10 before. See response to point 16 below for revised text (now on Page 11).

10. Misspelling “Cylindrical” p. 9

Thank you. We corrected it on what is now Page 11, reproduced below:

To avoid edge effect at the cut in the periodic dimension x_2 of the ~~cylindrical~~ cylindrical chart, we computed PIV on images tiled in x_2 .

11. Why is MATLAB used – would Python enable more accessible and open size and updates?

There are historic reasons for why we used MATLAB — an investment in the past that is more difficult to move away from in the modular platform we adopted. We agree that including a Python-based method of interacting with the atlas would improve accessibility. We have now modified the presentation of our atlas accordingly, and developed a new Python-based interface (<https://zenodo.org/records/14285126>) to interact with the atlas (via Jupyter notebook). In particular, we have now delineated two separate interfaces that are tailored for the two types of users we anticipate:

(1) users who primarily wish to access the atlas data as it currently exists (Python interface: <https://zenodo.org/records/14285126>)

(2) users who wish to add their own data to the atlas and analyze new ensembles of data (Matlab interface: <https://github.com/npmitchell/dynamicAtlas>)

For (1), we have compiled a master spreadsheet of the properties of the data currently in our atlas in a CSV file format (uploaded to our updated Data Dryad repository: <https://datadryad.org/stash/dataset/doi:10.25349/D9WW43>), and written a Python-based interface (located at: <https://zenodo.org/records/14285126>) to explore properties of these data using this CSV as a guide, via the interactive Jupyter notebook environment. We have included in the Supplementary Information (Page 30) a tutorial Jupyter notebook that enables the user to view or analyze data of interest (Tutorial also located at: <https://zenodo.org/records/14285126>, and CSV located at: <https://datadryad.org/stash/dataset/doi:10.25349/D9WW43>). We anticipate that most users will interact with the atlas using this method, and that it will be accessible at all levels of computational experience. We have successfully had our Python software tested on Mac OS Sequoia 15.1, Windows 10, and Linux Ubuntu 18.04.6, in the hands of users with limited Python experience and no prior experience of the software.

For (2), we have written functions in MATLAB to perform this analysis (located on the DynamicAtlas GitHub: <https://github.com/npmitchell/dynamicAtlas>). To improve accessibility, we have created a new detailed step-by-step tutorial (included as the file “DynamicAtlas_Demo_Tutorial.pdf” on the DynamicAtlas Github: <https://github.com/npmitchell/dynamicAtlas>), including screenshots, to guide the user through the entire pipeline of creating a consensus timeline, using a demo dataset (see response to point 13).

We built this code in MATLAB because that is the language that we have the most experience with, but nothing fundamental about the code functionality is unique to MATLAB. We also feel that MATLAB is a natural choice for our more computationally intensive methods, particularly for a broad acceptance by a biological audience, given the active support infrastructure and extensive library of documentation material provided by the developer MathWorks. Additionally, an

extensive ecosystem of existing scientific users and institutional licenses exists for MATLAB.

12. Why is 0.2 min timestep used for all temperatures?

Thank you for raising this point. We have now modified the text to clarify what this refers to. See additions to section 'Methods: Statistical comparison of embryos at varying temperatures' on Page 11, reproduced below:

In order to compare tissue deformations between temperature conditions, we first define the tissue displacement by integrating the coarse-grained tissue velocity fields over time using a Runge-Kutta 4th order scheme. For the purposes of numerical integration we linearly interpolate the velocity field (known at grid points) in space and time. We chose a numerical integration timestep of 0.2 minutes for smoothing the trajectories (the value was chosen such that multiple smoothing timesteps are integrated within our time resolution of 1 minute, but its value is arbitrary).

Our choice of an integration timestep of 0.2 min can be regarded analogous to a numerical smoothing parameter in time; our results are agnostic to this value. We used this timestep across temperature conditions to be consistent in our integration algorithm.

13. Enabling a demo with smaller dataset sizes would be useful for testing and evaluation of the approach from outside users.

We thank the reviewer for this feedback, and have incorporated this into the changes we have made in our code presentation. In particular, we have assembled a smaller demonstration dataset (located at <https://zenodo.org/doi/10.5281/zenodo.14792464>) that users can use to test our Matlab-based code (<https://github.com/npmitchell/dynamicAtlas>), including the time-alignment and time-stamping methods. In addition, we have created a new tutorial demonstration for our Python-based interface to query the atlas (<https://zenodo.org/records/14285126>).

14. Does the inclusion of the Runt dataset essential for building the consensus, or could it be done in the absence of gene expression landmarks?

We have changed the presentation of Figure 2 and Figure 6 to make the generality of the timeline creation methods more clear. Runt was used as a spatiotemporal

cue for aligning fixed and live datasets onto each other because the pattern of Runt expression changes in predictable ways over time that can be mapped onto a consensus timeline. This same approach could be done with any other gene expression pattern that changes in reproducible, distinguishable ways over time. However, as demonstrated in Figure 2D-D” and Figure 6, we also show the utility of our approach to build a consensus developmental timeline in the absence of gene expression patterns. We can use either tissue flow integration (Figure 2D-D”) or tissue geometry (Figure 6) to achieve this.

15. How would this framework work in another non-model system or stage without nice gene expression pattern data availability?

That is an excellent question. We have implemented our atlas methods in the *Drosophila* embryo context, but have constructed our atlas architecture for greater applicability. We have used gene expression data as a convenient means of aligning embryos in morphological time. But in other cases, we have used morphological features for the alignment (for example, midgut alignment which uses organ shape changes, see Figure 6). Our time-alignment methods do not require gene expression pattern availability.

Rather, our methods depend on a less restrictive condition: that an embryo exhibits a deterministic feature of development which changes with time reproducibly across samples (for example — but not limited to — gene expression patterns (Fig. 2A-B”), tissue deformations via integrated flows (Fig. 2D-D”), and organ geometries (Fig 6C)). To observe such dynamic features, we extend developmental biologists’ approach of classifying developmental stages, by using high time resolution movies to capture geometric changes between, and within, stages. Quantitatively, geometric changes are equivalent to integrated flows, so we can use our integrated flow method broadly to align live datasets in morphological time. Because reproducible shape changes characterize development across species, we expect this framework to function for a wide range of developmental biology problems, including but not limited to the morphogenesis of vertebrate and invertebrate embryos.

We have made additions to the main text accordingly. See addition to Section 3: ‘Aligning live datasets based on tissue deformation’ on Page 3, reproduced below:

Note that we use total tissue deformation, and not the instantaneous flow field, as a developmental landmark. In many contexts, instantaneous flow fields can be relatively constant in time (e.g. cells migrating with fixed speed), making them unsuitable as timestamps.

This method time aligns distinct embryos from tissue deformation

alone, and does not require analysis of gene expression patterns. It is possible that this method could be used in other living systems.

See also the addition to the Discussion on Page 8, reproduced below:

More broadly, our framework enables in toto spatiotemporal alignment of embryos using reproducible features of developmental dynamics. We hope that this work may serve as a template for quantitative analysis of morphogenesis in other living systems.

16. What are sample sizes in figure 4?

We have clarified both the main text and the legend for the figure (now Figure 5) accordingly. Revised on what is now Page 11, reproduced below:

For each condition, we have imaged at least three embryos ($n=3$ for 17°C , $n=5$ for 22°C , $n=3$ for 27°C). We can therefore compute the mean $\overline{\Delta_T}$ and variance σ_T^2 of Δ_T over samples i within each temperature condition.

These sample sizes were used for all panels in the figure.

17. A couple of terms that could be defined more simply: Radon transform (fig. 1), Pullback

In changing the legend for Figure 1, the term 'Radon transform' has been removed. We have modified references to the term pullback, see changes to section 'Methods: Quantification of tissue deformation with PIV' on Page 11, reproduced below:

This procedure was followed for every embryo in the ensemble, resulting in a spatially discrete velocity field $v_{\text{embryo}}(t, x_1, x_2)$ with a vector at each point of the lattice grid. Since all ~~pullbacks~~ 2D map projections were generated in the same coordinate space, all resultant velocity fields were directly comparable at corresponding coordinate locations between embryos.

We have clarified our definition of 'instantaneous flow', see additions to cross-correlation section on Page 3 and Page 4, reproduced below:

We measured *instantaneous* tissue flow (Fig. 3A) in all live datasets of the atlas using particle image velocimetry [32]. A key observable from the instantaneous flow is the flow pattern (i.e. flow normalized by its magnitude), which describes the directions cells move in from one moment to the next (see SI: Instantaneous Flow). When we analyze these ~~tissue flows~~ flow patterns, we find discrete periods of time in which the global pattern of tissue velocity *within an embryo* remains remarkably stationary (Fig. 3B).

We have also replaced the jargon of 'field' with the simpler term 'pattern' when introducing comparisons between flows. See addition to cross-correlation section on Page 4, reproduced below:

We quantify this observation by comparing the velocity ~~field~~ pattern at time t to the velocity ~~field~~ pattern at a later time t' , which defines a pairwise correlation function of tissue flow to itself. Computing pairwise correlations between flow patterns in time yields an autocorrelation matrix. We report the autocorrelation on a scale of -1 (perfectly anticorrelated) to 1 (perfectly correlated), shown in Fig. 3B for a movie of a wild-type embryo, recorded from the onset of gastrulation to dorsal closure. The magnitude of autocorrelation is high during discrete blocks of morphogenetic time, indicating that a modular sequence of flow patterns governs embryogenesis. We refer to these blocks as modules of stationary flow patterns.

Reviewer #3 (Remarks to the Author):

Lefebvre et al. present a morphological atlas that encompasses several stages of *Drosophila* embryonic development. Their atlas is based off of cutting-edge multi-view SPIM imaging that allows an in toto view of the fly embryo. The lab is also expert in segmentation and computational analysis which is performed through customized MatLab coding – these segmented data are then used to generate the atlas. The atlas is compiled from live imaging 495 different fly embryos in 19 different backgrounds (WT and 18 genetically compromised backgrounds). It represents a serious amount of work, and one of the more interesting aspects of their methodology are the approaches that are used to extract cell and tissue flows from so many different embryos that will each have variations in the total size and timing of the processes that occur in them. The study is nice in that they have established elegant solutions to these issues. This is clearly a methods-oriented paper, so, in this respect, is very appropriate for Nature Methods. As the authors acknowledge, it would have been more interesting if the atlas morphologies could have been tied back to changes in gene expression (e.g., RNAseq), but this is admittedly very difficult to do. The MatLab approaches are also fairly standard in the field, but applied smartly and to a high-level. I therefore think this study could still be interesting to the Nature Methods community. They also show the utility of such an atlas by observing how temperature changes differentially impact different processes in development (gastrulation movements vs cell mitoses), and also nicely show an example of deeper tissue morphological changes, which poses different challenges than surface tissue analyses. I did not find many of the conclusions surprising (outside of the differential impact temp had on mitosis vs morphologies, though why this is the case is unclear) however. The paper is well-written, and data is adequately displayed. I do not have too many critiques; it is largely a well-done study and dataset and to me it just comes down to impact and suitability for the journal. A few further comments...

1) Figures 1 and 2 are very similar flowcharts, it would be nice to try to differentiate these visually as well as provide more methodological details to distinguish them.

Thank you for this feedback. We have incorporated this into our new figure design. We have removed information from Figure 1, and have now re-constructed this figure as a big-picture graphical overview of the dynamic atlas and its methods. We have also changed Figure 2 to be clearly visually distinguishable from Figure 1, and made clear that Figure 2 now specifically depicts the different methods we have developed to align datasets together in time. We have added panels 2D-2D'' to Figure 2 as well, to emphasize that our alignment methodology does not require the analysis of specific gene expression patterns. Previously, the two figures both had

elements of a general overview and a presentation of specific methods. Now, Figure 1 just contains the former, while Figure 2 just contains the latter.

2) Standard reviewer complaint that n numbers and statistical tests used should be listed in each figure legend ;-)

Thank you. We have modified all figure legends accordingly to include descriptions of all sample sizes and statistical tests.

3) The authors seem to make a point of commenting that several tissue flows (like germband extension) are highly time dependent and reproducible. Is this interesting? Aren't there many tissues like this? Is there a contrast to be made?

Thank you for raising this point. We have made several new clarifications and additions to address it. There are two different kinds of reproducibility we explore in tissue flows: 1) embryo-to-embryo reproducibility, 2) reproducibility within an embryo. The first is well-established, however the second was to our knowledge not described. We find that flow patterns are stationary within the embryo, contrasting with the PRG patterns which continuously deform. To highlight this, we have added panels F through I to Figure 3, to directly illustrate the contrast between gene expression patterns and flow patterns during germband extension. Such flows — with both a constant derivative and continuously-changing positions of the material — are common in non-living systems, but those systems do not match the complexity of biology. Here, we find in a biological context that the germband extension flow, at the level of position, is time dependent, yet critical aspects of its flow are not.

We clarify this point in the main text accordingly. See addition to the Introduction on Page 1, discussing the context of PRG patterns and flow patterns:

*A major challenge in all atlases is to understand the embryo's dynamics, by which gene expression patterns deform over time. For example, in *Drosophila*, PRGs are known to be important drivers of germband extension, the phase of gastrulation when the body axis elongates [17]. While the BDTNP atlas spatially maps early PRG patterns, the atlas's coarse time alignment technique precludes detailed characterization of their dynamic timecourse, and the relationship between PRG patterns and tissue flow patterns remains elusive. To understand these dynamics how gene expression patterns in an animal regulate its shape changes, one requires an atlas with live in toto movies, in order to directly study the dynamics of*

~~the~~ *its* tissue motion.

Further, see added text about flow patterns on Page 3 and Page 4, reproduced below:

We measured *instantaneous* tissue flow (Fig. 3A) in all live datasets of the atlas using particle image velocimetry [32]. A key observable from the *instantaneous flow* is the flow pattern (i.e. flow normalized by its magnitude), which describes the directions cells move in from one moment to the next (see SI: Instantaneous Flow). When we analyze these ~~tissue flows~~ *flow patterns*, we find discrete periods of time in which the global pattern of tissue velocity *within an embryo* remains remarkably stationary (Fig. 3B). We stress that the tissue itself is not stationary, but instead the pattern of motion is stationary: although the cells are moving across the embryo, the ~~pattern generated by the flow~~ *flow pattern* is stationary during certain discrete stages of development. (We call these flows ‘quasi-stationary’ — as their magnitudes may vary — and call the flow patterns ‘stationary’.)

Also see added text about pair-rule gene expression pattern correlations on Page 4, reproduced below:

PRGs are important contributors to GBE [17]. Yet, the *in toto* pattern of PRGs has only been explored in fixed samples. The atlas enables quantitative analysis of PRG pattern kinematics, shown for a set of six PRGs in Fig. 3F-H. Using the middle time for DC and GBE modules, we quantify the autocorrelation patterns of both PRGs and flows (Fig. 3G,G',H). Correlations of PRGs and flows are both high during DC. In contrast, during GBE, PRG autocorrelation drops within 2 minutes. This fast PRG autocorrelation drop can be quantitatively explained from vorticity (Fig. 3F'), and speed increase during GBE (Fig. 3I). PRG expression becomes reshaped as tissue vorticity increases, and within 2 minutes, stripes no longer overlap faithfully (see Methods: Pair-Rule Gene De-Correlation). This fast de-correlation from flow suggests that PRGs no longer directly instruct GBE shortly after the flow pattern has been established.

We have further augmented the Discussion section to discuss the significance of this contrast. See additions to the Discussion on Page 7, reproduced below:

Quasi-stationary flows in the early Drosophila embryo provide a simple physical way to establish complex shape changes. Despite the complexity of the embryo, with order 10, 000 cells and 10, 000 genes [51], we find that the embryo executes the same flow pattern for an extended time. This stationary flow pattern is established by active processes in cells, but it does not change while cells move through it. Instead, the flow rapidly changes once the tissue has achieved a desired geometry.

The adherence of ~~tissue flow~~ the GBE flow pattern to a fixed, geometric frame is reminiscent of the well-documented spatial precision of gene expression patterns in the early blastoderm [52, 53]. These earlier studies showed gene expression levels to be spatially patterned with single-cell precision along the body axes. Tissue kinematics are likewise strongly correlated in a fixed, geometric (Eulerian) reference frame the fixed reference frame of the egg, despite massive motion of the material. This presents a challenge: how does genetic signaling in cells instruct the directions of flow, given that the flow pattern remains aligned with the principal axes of the egg, while cell positions — and thereby expression patterns of nuclear gene concentration — continuously change? The global distribution of myosin is a predictor of tissue flow during GBE [24]. Stationary GBE flow patterns suggest that myosin patterns are also stationary in the body axis frame. This result is in line with the recent finding that the global myosin pattern appears controlled by static cues that align with the DV axis during GBE tissue motion [30]. At the cellular level, recent work has showed cell-cell interfaces preferentially replenish their myosin level according to their degree of DV axis alignment [54]. We find that DV, but not AP patterning is required for stationary flow patterns. Taken together, this adds to the growing body of evidence that the DV patterning system coordinates robust morphogenetic movements of GBE, e.g. by patterning mechanical feedback [55] [56].

Additionally, to gain further understanding into the novelty of the scaling during GBE, we have conducted perturbations of the germband retraction (GBR) module under different temperature conditions (21°C, 23°C, 25°C, 27°C, n=1 for each condition). We have

included a new supplementary figure, SI Fig. S12, where we demonstrate that the flow rates during GBR do not scale with temperature. This is in contrast to GBE, illustrating that the scaling we find during GBE does not occur in all modules. We have added text to the temperature section on Page 6 accordingly:

In contrast, the GBR module does not scale with temperature (SI Fig. S12). Notably, unlike GBE, GBR is not characterized by cell intercalations in the germband, but by mechanical coupling to the amnioserosa [41][42].

DynamicAtlas: A Morphodynamic Atlas for Drosophila Development

(NMETH-RS57980A)

Point-by-Point Response, Second-Round Revisions

Editor:

Our point-by-point response to remaining reviewer comments follows below, with our responses in magenta.

Reviewer #1 (Remarks to the Author):

This paper describes computational methods for integrating fixed and live imaging data into a single atlas, using morphological data. Additionally, this paper describes a very useful resource resulting from these methods: an atlas of *Drosophila* germband extension that incorporates both gene expression data and the dynamics tissue rearrangement, designed as an open source tool to which future datasets can be added. Both the technique and resource described are novel and provide a significant advance to the field. My concern with the previous draft of this manuscript was that I was unable to run the code with the information provided. This concern has been amply addressed: the authors now provide a comprehensive and easy-to-follow tutorial for their Matlab code, as well as a new, simpler Python interface to access the current version of the atlas. These are major improvements, and should make their work broadly accessible to people in the field.

We thank the reviewer for the positive overall feedback, and we are especially glad that our changes have made the atlas software more accessible.

Reviewer #2 (Remarks to the Author):

The work of Lefebvre et al. addressed many points raised in the first review round, which has improved the work. The manuscript also provides a clearer description of interesting insights gained from the quantitative approaches developed.

The work is innovative in its analysis of morphogenesis.

The present work shows that there are differences in scaling with temperature between flow velocity and mitotic times. Second, the work illustrates how patterning directs future modules of morphogenesis at discrete time points, which are characterized by quasi-stationary flow during GBE. This phenomenon depends on DV patterning, suggesting that cross-regulatory principles define constraints on modules. The analysis is then generalized to a second example of 3D tubular morphogenesis.

Question:

For SI Table 1 & 2, can columns be added to clarify if they were used in a specific figure panel or video in the manuscript? The relationship between the resources and data is not clear.

Remarks on code availability:

The SI provides details of tutorial, which aids in understanding the workflow of the code.

We thank the reviewer for raising the question about the relationship between resources and data. In response, we have modified SI Tables 1 & 2 accordingly, by adding an extra column to each to clarify which resources were used in specific figure panels / supplementary videos.

We thank the reviewer for the positive feedback on both the resource and the code availability.

Reviewer #3 (Remarks to the Author):

I have no further concerns -- the authors have responded appropriately to reviewer concerns and seriously addressed the critiques. The revised manuscript seems like a good addition to Nature Methods.

We thank the reviewer for the positive overall feedback.